



# A random forest model to assess snow instability from simulated snow stratigraphy

Stephanie Mayer[1], Alec van Herwijnen[1], Frank Techel[1], and Jürg Schweizer[1]

[1]WSL Institute for Snow and Avalanche Research SLF, Davos, Switzerland

**Correspondence:** Stephanie Mayer (stephanie.mayer@slf.ch)

**Abstract.** Modeled snow stratigraphy and instability data are a promising source of information for avalanche forecasting. While instability indices describing the mechanical processes of dry-snow avalanche release have been implemented into snow cover models, there exists no readily applicable method that combines these metrics to predict snow instability. We therefore trained a random forest (RF) classification model to assess snow instability from snow stratigraphy simulated with SNOWPACK. To do so, we manually compared 742 observed snow profiles with their simulated counterparts to select the simulated weak layer corresponding to the observed rutschblock failure layer. We then used the observed stability test result and an estimate of the local avalanche danger to construct a binary target variable (stable vs. unstable) and considered 34 features describing the simulated weak layer and the overlying slab as potential explanatory variables. The final RF classifier aggregates six of these features into the output probability $P_{\text{unstable}}$, corresponding to the mean vote of an ensemble of 400 classification trees. Although the training data only consisted of 146 manual profiles labeled as either unstable or stable, the model classified profiles from an independent validation data set with high reliability (accuracy: 88%, precision: 96%, recall: 85%) using manually predefined weak layers. Model performance was even higher (accuracy: 93%, precision: 96%, recall: 92%), when the weakest layers of the profiles were identified with the maximum of $P_{\text{unstable}}$. Finally, we compared model predictions to observed avalanche activity in the region of Davos for five winter seasons. In 73% of the days, our model correctly discriminated between avalanche days and non-avalanche days. Overall, the results of our RF classification are very encouraging, suggesting it could be of great value for operational avalanche forecasting.

## 1 Introduction

Forecasting snow avalanches in mountainous terrain has long proved to be a challenge for researchers and operational forecasters. The probability of avalanche release depends on snow instability, the sensitivity of the local snowpack to artificial or natural triggers (Statham et al., 2018). Snow instability results from a complex interplay between snowpack, terrain and various meteorological drivers over time (e.g. Schweizer et al., 2003a; Reuter et al., 2015b). To estimate snow instability at a specific location, stability tests such as the rutschblock test (RB) or the Extended Column test (ECT) can be performed (e.g. Schweizer and Jamieson, 2010; Techel et al., 2020b). Snow stability tests consist of incrementally loading an isolated block of snow of pre-defined dimensions to evaluate the load required to fracture weak layers in the snowpack. Such observations of snow instability constitute an essential source of information for the preparation of avalanche forecasts intended to warn the





public about the avalanche danger. However, snow instability measurements are very time-consuming, sometimes dangerous to perform and only provide very local information for one point in time. Although the potential of numerical snow cover models to increase the spatial and temporal resolution of snow instability data has been recognized, operational avalanche forecasting rarely incorporates modeled snow instability data (Morin et al., 2020).

A major reason for the limited use of modeled snow instability data in avalanche forecasting is the complexity of the processes involved in avalanche formation. The release of a dry-snow slab avalanche is a fracture mechanical process which starts with failure initiation in a weak layer below a cohesive slab followed by subsequent rapid crack propagation across the slope (e.g. Schweizer et al., 2003a; van Herwijnen and Jamieson, 2007; Gaume et al., 2017). Modeling snow instability thus requires (i) modeling snow stratigraphy including relevant weak layers, (ii) a suitable choice of parameters describing

the relevant mechanical processes and (iii) a meaningful interpretation of these parameters. Modeling the one-dimensional snow stratigraphy (step i) is feasible with the two most advanced numerical snow cover models CROCUS (Brun et al., 1989; Brun et al., 1992; Vionnet et al., 2012) and SNOWPACK (Lehning et al., 1999; Bartelt and Lehning, 2002; Lehning et al., 2002a, b). Both physically-based models are driven with meteorological data from either automatic weather stations or numerical weather prediction models (e.g. Bellaire and Jamieson, 2013; Quéno et al., 2016) and provide microstructural (e.g. grain

size) and macroscopic properties (e.g. density) for each snow layer. Several validation campaigns demonstrated a reasonably good agreement between modeled and observed snow stratigraphy (e.g. Durand et al., 1999; Lehning et al., 2001; Monti et al., 2009; Calonne et al., 2020), and, in particular, confirmed the models capability to reproduce critical snow layers such as surface hoar (Bellaire and Jamieson, 2013; Horton et al., 2014; Viallon-Galinier et al., 2020). From the basic model output, different mechanical properties can be calculated. SNOWPACK contains a module for mechanical stability diagnostics which includes

various parameters describing the processes of avalanche formation. To assess dry snow instability, a potential weak layer is determined with the structural stability index (SSI, Schweizer et al., 2006) or the threshold sum approach (Monti et al., 2014), and stability indices are then calculated for this layer. The skier stability index (SK38) describes failure initiation (Föhn, 1987b; Jamieson and Johnston, 1998; Monti et al., 2016) and the recently implemented critical cut length ($r_c$) relates to crack propagation (Gaume et al., 2017; Richter et al., 2019). While SK38 and $r_c$ should capture the most important processes involved in

the formation of human-triggered avalanches (step ii), the interpretation of these stability indices (step iii), however, remains a major challenge. Although both indices were related to avalanche observations or signs of instability in several field studies using observed snow properties (e.g. Jamieson and Johnston, 1998; Gauthier and Jamieson, 2008; Reuter and Schweizer, 2018), there are only few validation studies based on simulated snow stratigraphy (e.g. Schweizer et al., 2006; Richter et al., 2019). In particular, there are no validated threshold values for a combination of both indices in the case of simulated snow profiles.

Moreover, the SK38 provides meaningful results only for weak layers that are not deeply buried (<80 cm) (Schweizer et al., 2016; Richter et al., 2021).

Given the limitations of the process-based snow instability indices, we aim at assessing dry-snow instability from simulated snow stratigraphy employing a machine learning approach. Our goal is to develop a model which aggregates information on snow stratigraphy into a probability of instability provided for each layer of the simulated snow profile. This model should

offer the possibility of detecting the weakest layer of a snow profile and assessing its degree of instability with one single



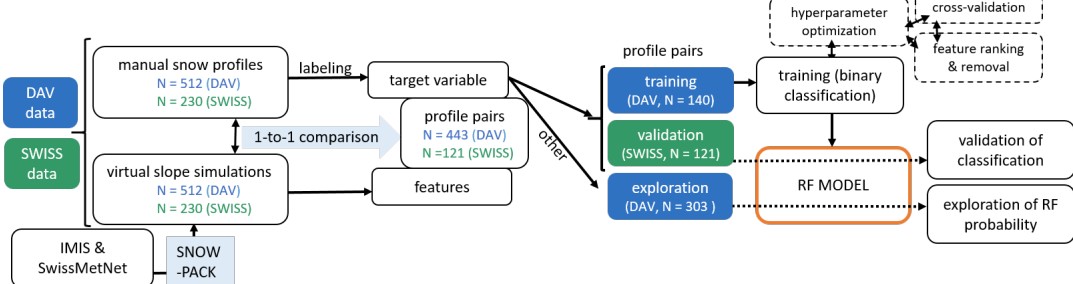

**Figure 1.** Overview of data sets and steps used for data pre-processing (left side) and model training and evaluation (right side).

index. To this end, we incorporate existing stability indices as well as microstructural and macroscopic snow layer properties as input variables into a Random Forest (RF) classification model. We construct this model using a one-to-one comparison of SNOWPACK simulations with observed snow profiles including stability test results. To discriminate rather unstable from rather stable snow conditions, we derive threshold values for the predicted probability of instability. A comparison of modeled

snow instability with observed avalanche activity highlights the potential of our model to assess snow instability.

## 2 Data and data preparation

Our approach to classify snow profiles using simulated snow stratigraphy was divided into several steps (Figure 1), which involved the use of two data sets (DAV and SWISS). Each of the data sets consisted of pairs of observed snow profiles (Sect. 2.1) and associated simulations at virtual slopes which were obtained by using meteorological forcing data (Sect. 2.2) as input

for the SNOWPACK model (Sect. 2.3). Pre-processing the data included a one-to-one comparison of manual and simulated snow profiles. We manually defined a weak layer in the SNOWPACK simulations corresponding to the rutschblock failure layer observed in the field and then discarded all profile pairs that did not meet predefined similarity criteria (Sect. 3.1.2). After these preprocessing steps, the DAV data set was used to train the classification model (Sect. 3.1.3), which was then validated on the SWISS data set (Sect. 3.2).

### 2.1 Manual snow profiles and stability observations

### 2.1.1 DAV data set

To train our classification model, we used snow profiles observed in the region of Davos (Eastern Swiss Alps, Switzerland; Appendix A: Figures A1 and A2) from the 18 winter seasons of 2001-2002 to 2018-2019 (data set used by Schweizer et al., 2021b, accessible at Schweizer et al. (2021a)). This data set (DAV), consisted of 512 profiles containing information on the

profile site (coordinates, slope angle, slope aspect), snow stratigraphy (grain type and size, snow hardness index) observed according to the ICSSG (Fierz et al., 2009), a rutschblock test (RB, Föhn, 1987a; Schweizer and Jamieson, 2010), and an estimate of the local avalanche danger level (local nowcast: LN; Techel and Schweizer, 2017). The RB test result included the




rutschblock score (ranging from 1 to 7), the height of the failure interface, release type (whole block, partial release below skis or only an edge) and quality of the fracture plane (smooth, rough, irregular). The local nowcast assessment of avalanche danger

is provided by researchers, forecasters or specifically trained observers, using a five-level danger scale (1-low, 2-moderate, 3-considerable, 4-high, 5-very high). The assessment refers to the area observed during a day traveling in the backcountry, which is typically at the order of several square kilometers, and does not refer to a single slope (for more details refer to Techel and Schweizer, 2017). The mean snow depth of all profiles in the DAV data set was 111 cm and 64 % of the RB tests failed adjacent to a layer of persistent grain type.

To evaluate the model, we also used visual observations of dry-snow avalanches from the region of Davos for the five winter seasons 2014-2015 until 2018-2019 (data set used by Schweizer et al. (2020a) and accessible at Schweizer et al. (2020b)). From this data set, we extracted dry-snow avalanches which either released naturally, were human-triggered or had an unknown trigger type. These data were aggregated into the avalanche activity index (AAI), a weighted sum of all observed avalanches on a specific day, with weights assigned according to avalanche size (weights 0.01, 0.1, 1, and 10 for size classes 1 to 4,

respectively; Schweizer et al., 2003b) and type of triggering (weights 1, 0.81, and 0.5 for trigger types "natural", "unknown", "human", respectively; Schweizer et al., 2020a). We further defined an avalanche day as a day with at least one recorded avalanche of size class 2 or greater.

### 2.1.2 SWISS data set

For model validation, we compiled an independent data set of 230 snow profiles (SWISS, Figure A1), again including a RB

test and a LN assessment of the avalanche danger level. These profiles were observed at various locations throughout the Swiss Alps, not including the region of Davos, during the winter seasons of 2001-2002 to 2018-2019. To perform representative SNOWPACK simulations, we selected profiles which were observed within a horizontal distance of 10 km and a vertical distance of 200 m of an automated weather station (AWS). Moreover, the data recorded by the corresponding AWS could not have gaps of more than 24 hours. The mean snow depth of the SWISS profiles was 138 cm and 45 % of the RB failure

interfaces were located adjacent to a layer of persistent grain type.

### 2.2 Meteorological forcing data

To simulate the snow cover at the sites of the observed snow profiles, we forced SNOWPACK with meteorological data from a network of automated weather stations (AWS) located between 1500 m and 3000 m a.s.l. across the Swiss Alps (Intercantonal Measurement and Information System: IMIS; Lehning et al., 1999). These IMIS stations are located at mostly flat sites consid-

ered representative of the surrounding area. Meteorological variables and snow cover properties were recorded every 30 min, and included air temperature (non-ventilated), relative humidity, wind speed and direction, reflected shortwave radiation, snow surface temperature and snow height. In addition, the majority of the AWS are equipped with a rain gauge which is unheated, and thus do not provide reliable measurements of solid precipitation. For the simulations of the DAV data set, we also used data from two SwissMetNet stations, operated by the Swiss Federal Office of Meteorology and Climatology (MeteoSwiss) and





a research station, operated by SLF. These stations also measure incoming short- and long-wave radiation with ventilated and
heated sensors as well as solid and liquid precipitation with heated rain gauges.

## 2.3 SNOWPACK setup

### 2.3.1 DAV data set

For the SNOWPACK simulations in the DAV data set, we interpolated measurements from six AWS from the IMIS network
(WFJ2, DAV2, DAV3, KLO2, KLO3, SLF2), two SwissMetNet stations (WFJ, DAV) and the research station STB2 to the
exact locations of the individual snow profiles. For the locations of snow profiles and AWS see Appendix A (Figures A1 and
A2). In addition to the precipitation measurements from WFJ and DAV, we also used estimated precipitation values for the five
IMIS stations obtained from SNOWPACK runs driven with measured snow depth (Lehning et al., 2002a; Wever et al., 2015),
employing an empirical relationship for new snow density as a function of air temperature, relative humidity and wind speed
(Schmucki et al., 2014).

To spatially interpolate meteorological parameters to the exact locations of the manual snow profiles, we used the pre-
processing library MeteoIO which applies a combination of lapse rate and inverse distance weighting (Bavay and Egger,
2014).

Starting at 1 October of the respective winter season, each simulation was run at the location of the manual snow profile up
to the exact date and time ($\pm 0.5$ h) when the manual profile was observed, using a time step size of 15 min. To account for
slope angle and aspect, the simulations were carried out for so-called virtual slopes, i.e. short-wave radiation and precipitation
amounts were projected onto the slope, while other influences of surrounding terrain on the snowpack were neglected. Energy
fluxes at the snow-atmosphere boundary were calculated using Neumann boundary conditions. For the soil heat flux at the
bottom of the snowpack we employed a constant value of 0.06 $\mathrm{W/m^2}$ (Davies and Davies, 2010). The flow of liquid water
through the snow cover was modeled applying Richards equation (Wever et al., 2014). Finally, to obtain a simulated snow
depth close to that of the manual snow profile, we scaled interpolated precipitation values as

$$P_{\mathrm{corr},i}(t) = \frac{\mathrm{HS}_{\mathrm{obs},i}}{\mathrm{HS}_{1,i}} P_{1,i}(t), \tag{1}$$

where $P_{\mathrm{corr},i}(t)$ is the scaled precipitation for profile $i$ at time step t, $\mathrm{HS}_{\mathrm{obs},i}$ is the snow depth observed at the manual profile
$i$, $\mathrm{HS}_{1,i}$ is the simulated snow depth from a first unscaled SNOWPACK run for the same location and time as the observed
profile, and $P_{1,i}(t)$ is the interpolated unscaled precipitation used in the first simulation. Each simulation was then re-run using
the corrected precipitation $P_{\mathrm{corr},i}(t)$ to drive snow accumulation.

### 2.3.2 SWISS data set

For the SWISS data set, we simulated snow stratigraphy at the location of the nearest IMIS station (i.e., in contrast to the DAV
data set, we did not interpolate meteorological data to the exact profile location). As for the DAV data set, we performed virtual
slope simulations with slope angle and aspect corresponding to the manual profile. The measured snow surface temperature



was imposed as Dirichlet-type upper boundary condition for the energy exchange at the snow surface. To ensure that the energy input was not underestimated during ablation periods, the upper boundary condition switched to Neumann-type (i.e. energy fluxes at the snow surface were calculated), whenever the snow surface temperature exceeded -1.0 °C. All further settings, including the scaling of precipitation, were identical to those of the DAV simulations.

## 3 Methods

An overview of the different steps and data subsets involved in the development (Sect. 3.1) and evaluation (Sect. 3.2) of our classification model is shown on the right-hand side of Figure 1.

### 3.1 Model development and optimization

#### 3.1.1 Target variable and features

The construction of the classification model required the definition of a meaningful target variable based on the observed stability. To this end, we labeled the manual snow profiles based on the results from the RB test and the LN assessment of avalanche danger. The snow profile and the RB test provide information on weak layers and their stability at the location of the snowpit. The location of these snow pits is often rather specific, as observers aim to find locations where snowpack stability is poor (e.g. McClung, 2002). The nowcast assessment, on the other hand, also considers observations at a larger scale, such as
recent avalanches, avalanche size and signs of instability (Schweizer et al., 2021b).

Based on the combination of RB score and release type, we grouped RB test results into three different stability classes *poor*, *fair* and *good* (Fig. 2a). While our approach is similar to that of Techel et al. (2020b), we only defined three classes by merging the two lowest classes *very poor* and *poor* of Techel et al. (2020b) into one class (*poor*). Besides this RB stability rating, we also considered LN as a second criterion to identify those profiles which were presumably most representative for
snow stability in the region and hence best suited for building the classification model. The frequency of the different stability classes varies with the danger level (Techel et al., 2020a). If a stability test belongs to a minority class at a given danger level, the simulated snow stratigraphy will likely not be able to reproduce the snowpack at that test location. In general, we cannot expect that the simulated snowpack can fully reproduce the snow depth, stratigraphy and stability as observed at the location of the manual snow profile, since we relied on interpolations of meteorological data to drive the 1 D simulations (see Sect. 2.3),
which, for instance, do not consider snow redistribution by wind in complex terrain. Therefore, we combined the RB stability classification and the LN assessment and assigned all snow profiles to nine different subgroups (Fig. 2b), of which only two were used to train the classification model. In the following, we denote the upper left and lower right of this 3x3 RB-LN-grid as *stable* and *unstable* class respectively, i.e.

$$\text{stable class} := \quad \{(\text{RB result} = \text{good}) \quad \text{and} \quad (\text{LN} = 1)\} \tag{2}$$

$$\text{unstable class} := \quad \{(\text{RB result} = \text{poor}) \quad \text{and} \quad (\text{LN} \geq 3)\}. \tag{3}$$





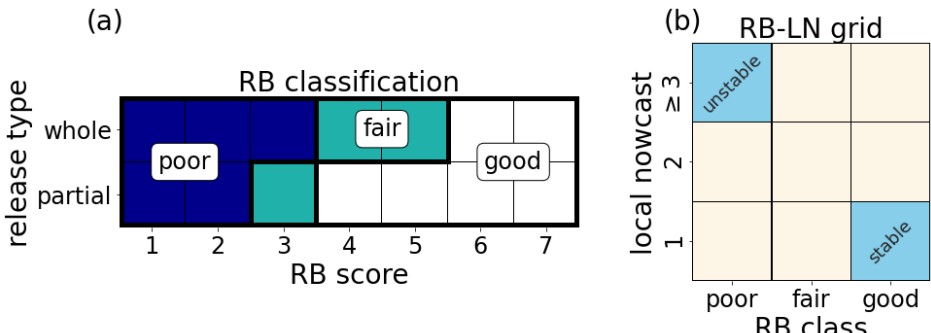

**Figure 2.** (a): Definition of the RB stability classes *poor*, *fair* and *good* in dependence on RB score and release type. (b): Classification of profiles into nine classes of a rutschblock stability - local nowcast (RB-LN) grid. The classification model was trained on DAV profiles belonging to either the upper left or the lower right classes (in blue).

For these two «extreme» classes in the corners, we hypothesize that it is more likely that the simulated snow stratigraphy, obtained with interpolated regional meteorological data, and the manual snow profile are similar, as compared to all other classes. The two classes *stable* and *unstable* thus constituted the binary target variable of our classification model. To train the classification model, we only used the subset of profiles from the DAV data set that belonged to these two classes. The remaining classes were used for model evaluation beyond binary classification. The SWISS data set, intended to validate the binary classification model, only contained profiles labeled as either *stable* or *unstable*.

A careful selection and creation of discriminant features is crucial to the predictive performance of any classification task (Duboue, 2020). For our classification model, we extracted features from the simulated snow stratigraphy describing the weak layer and the overlying slab. Overall, we used 34 features (see Appendix B: Table B1), either direct output from SNOWPACK, such as macroscopic (e.g. density) or microscopic (e.g. grain size) layer properties, mechanical properties (e.g. shear strength), stability indices (e.g. SK38), or derived properties (e.g. skier penetration depth) and variables constructed on the basis of expert knowledge (e.g. the ratio of the mean slab density and the mean slab grain size).

### 3.1.2 Profile comparison

For each simulated profile (DAV and SWISS), we manually selected the layer corresponding to the RB failure layer observed in the manual snow profile. This was done by visually identifying a weak layer in the simulations with similar grain type and hardness as the observed RB failure layer, taking into account the overall sequence of layers. Prominent hardness differences and layers consisting of depth hoar, surface hoar or crusts generally facilitated the subjective profile alignment (some examples in Figure 3). For an unstable profile pair, we always searched for a layer with properties characteristic of a typical failure layer (large grain size, low density, persistent grain type, etc.; c.f. Schweizer and Jamieson, 2003). As simulated snow profiles generally consist of more layers than manual profiles, we chose the layer with lowest density and largest grain size within the potential layers to define the weak layer. If the weak layer of the manual profile was not present in the simulation, we picked





an alternative weak layer within the modeled profile that corresponded best. For instance, in the profile pair in Figure 3a, we selected the depth hoar layer just below the slab in the simulated profile since the simulation did not contain a faceted layer below a crust as in the observed profile. For stable profile pairs, it was often not possible to find a layer with the same grain type and hardness as the RB failure layer. In that case, we chose a layer with similar properties as the observed layer (e.g. Figure 3b), rather than selecting a layer with typical weak layer properties. Clearly, the described matching approach is rather subjective and does not lead to an unambiguous choice of the weak layer in the simulated profile. To reduce subjectivity in the comparison of profiles, we adhered to the following criteria:

1. Difference in observed and simulated snow depth must not exceed 20 cm.

2. Simulated slab thickness must not deviate more than 20 cm from the observed slab thickness.

3. The difference in hand hardness index between the observed and simulated weak layer must not exceed 1 step.

4. Differences in observed and simulated mean slab hardness must not exceed 1 step.

5. The grain type in observed and simulated weak layer must either be both persistent (i.e. facets, surface hoar or depth hoar) or non-persistent.

A pair of profiles was included if both criteria 1 and 2, and at least two out of the three criteria 3 to 5 were fulfilled. For the profiles where the RB test did not release at all (i.e. RB score 7) and there was no estimate for the weakest layer, we judged the similarity of the profile pair comparing snow stratigraphy and selected a weakest layer in the simulation based on expert knowledge.

Applying the similarity criteria to the 512 DAV profile pairs, led to the exclusion of 69 profile pairs (13%). The number of profiles in the unstable and stable class of the DAV data set reduced to $N = 73$ and $N = 67$, respectively (Figure 4a). To obtain a balanced training data set, we included three additional layers for each of the two stable profiles with no RB failure (i.e. RB score 7). Of the 230 SWISS profiles, 121 profiles fulfilled the similarity criteria (53%); 75 were labeled as unstable, and 46 as stable (Figure 4b).

### 3.1.3 Training the classification model

We trained a Random Forest (RF) model (Breiman, 2001a) to distinguish between the stable and unstable profile classes of the DAV data set ($N = 73$ each), using the Python library scikit-learn (Pedregosa et al., 2012). We chose a RF model for this classification task, as, in contrast to parametric approaches and threshold-based methods, this model allows accounting for complex mutual dependencies between features without any pre-assumptions on the multi-variable relationship between observed stability and simulated stratigraphy (Breiman, 2001b). The RF model is a supervised machine learning algorithm which constructs an ensemble of decision trees for data classification. The average of the predictions from the individual decision trees yields the final prediction of the RF, where the probability for a given class is determined by the proportion of





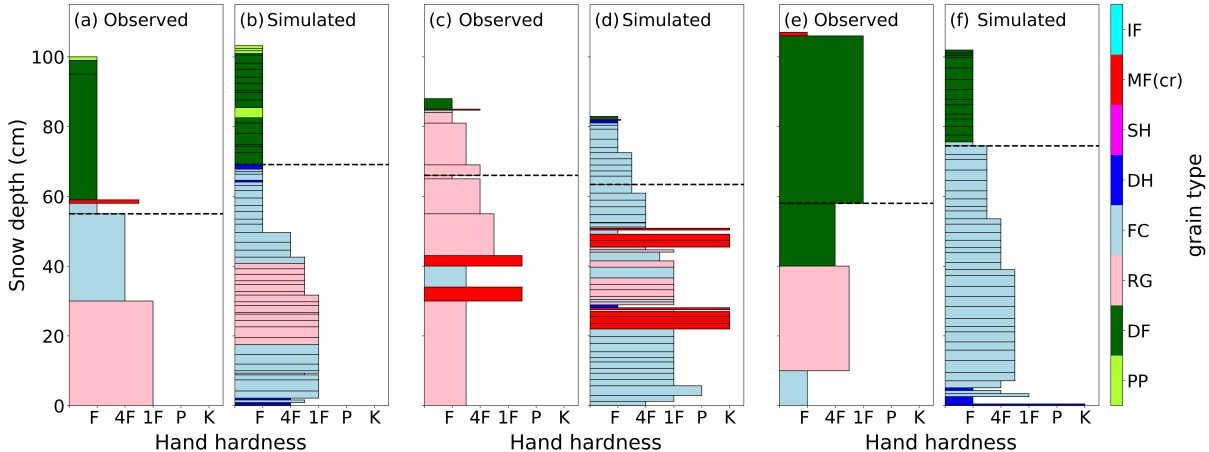

**Figure 3.** Exemplary profile pairs from the DAV data set with (a, c, e) manually observed snow profiles and (b, d, f) corresponding simulated snow stratigraphy from SNOWPACK. Hand hardness and grain type (colors) were coded after Fierz et al. (2009), where F corresponds to fist, 4F to four fingers, 1F to one finger, P to pencil, and K to knife. Grain types are precipitation particles (PP), decomposing and fragmented precipitation particles (DF), rounded grains (RG), faceted crystals (FC), depth hoar (DH), surface hoar (SH), melt forms (MF), melt-freeze crusts (MFcr) and ice formations (IF). The dashed horizontal line in the manual profile displays the observed height of the RB failure layer and the corresponding manually determined layer is indicated by a dashed line in the respective simulated profile. Profile pairs (a)-(b) and (c)-(d) passed the similarity check, while profile pair (e)-(f) was sorted out. Observed rutschblock results were (a) RB score 1, whole block, (b) RB score 5, edge and (c) RB score 4, whole block. The local nowcasts were given by (a) LN = 3, (b) LN = 1 and (c) LN = 2.

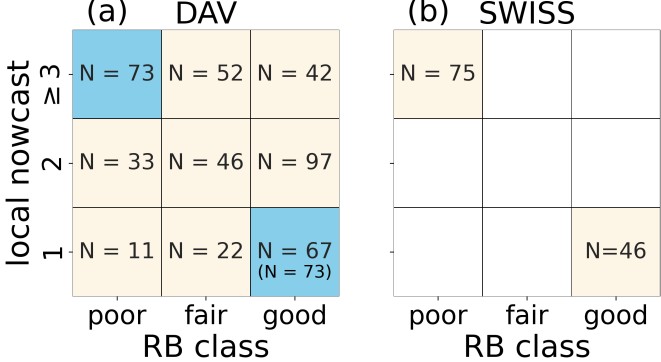

**Figure 4.** Classification of profiles for (a) the DAV data set and (b) the SWISS data set into RB stability - local nowcast classes. The numbers in the boxes denote the number of profile pairs per class which fulfilled the similarity criteria. The profiles in the blue boxes were used for the training of the classification model, while the beige boxes were used for model evaluation. The second number in the lower right class, $N = 73$, indicates that we included six additional layers of the stable profiles with RB score 7 to obtain a balanced training data set.





trees that voted for that class. Compared to a single decision tree, a RF estimator is less prone to overfitting to the training set, as its construction contains several sources of randomness (e.g. bootstrap sampling).

As the variety of split rules used within the ensemble of trees cannot be grasped by the human brain, RF can be considered as a black box-type classifier. Nevertheless, the RF algorithm includes a built-in feature importance estimation, based on evaluating the Gini impurity decrease at each split for every tree in the forest. The importance of a feature is computed as the normalized total decrease in Gini impurity brought by that feature within the ensemble of trees. The more a feature reduces the impurity, the more important the feature is.

The RF model includes several hyperparameters that can be optimized externally in order to customize the model to the data set of interest, in particular to prevent overfitting. The main hyperparameters include:

- the number of trees in the forest

- the maximal depth of a tree, i.e. the longest path between the root node and the leaf node

- the maximal number of features to consider for the best split

- the minimum number of samples required to split a node

- the function to measure the quality of a split

We optimized the hyperparameters before training the final RF model by systematically considering different hyperparameter combinations in a cross-validated grid-search. For every combination of hyperparameter settings, we trained a random forest model on five different subgroups of the training data set, and evaluated model accuracy on the left-out data. To prevent similar profiles being used for training and evaluation, we sorted the profiles by date before splitting the data set. We repeated the hyperparameter optimization process with different subsets of the complete set of features, avoiding highly correlated (Pearson's $r > 0.8$) pairs of features. Finally, we selected the combination of hyperparameters and feature subset which yielded the highest mean accuracy score (i.e. the ratio of correct predictions among all predictions) in the five-fold cross-validation.

Based on the feature importance ranking of the RF model with the optimized hyperparameters, we selected a subset of features with the highest ranking (feature importance $> 0.05$). We then conducted another round of hyperparameter optimization with the new choice of features ($N = 6$) and trained the final RF model with the optimized hyperparameters on the complete set of training data.

## 3.2 Model evaluation

### 3.2.1 Classifier performance on the SWISS data set

To evaluate the performance of the final RF model, we compared predicted and observed stability classes using the SWISS data set and standard performance measures based on a 2x2 contingency table (Figure 5) (Wilks, 2011). With the definitions shown in the contingency table, the accuracy, precision (positive predictive value), recall (true positive rate or sensitivity) and



| | | **Observed** | |
|---|---|---|---|
| | | **unstable** | **stable** |
| **Predicted** | **unstable** | True positive (TP) | False positive (FP) |
| | **stable** | False negative (FN) | True negative (TN) |
| | **total** | P | N |

**Figure 5.** Contingency table

specificity (true negative rate) are defined as:

$$\text{accuracy} := \frac{TP+TN}{P+N} \tag{4}$$

$$\text{precision} := \frac{TP}{TP+FP} \tag{5}$$

$$\text{recall} := \frac{TP}{P} \tag{6}$$

$$\text{specificity} := \frac{TN}{N} \tag{7}$$

To optimize the classification performance, we analyzed the receiver operating characteristic (ROC) curve, which describes the trade-off between recall and specificity along different classification thresholds discriminating stable from unstable profiles (Fawcett, 2006). The ROC curve is a diagnostic plot and is obtained by plotting the recall against the false positive rate (false positive rate $:= \frac{FP}{N} = 1 - \text{specificity}$) for different classification thresholds.

A random classifier would yield a diagonal line from $[0,0]$ to $[1,1]$, and a perfect model would be indicated by a ROC curve rising vertically from $[0,0]$ to $[0,1]$ and then horizontally to $[1,1]$. The area under the ROC curve (AUC) provides a metric to summarize the overall performance of a model with a value between 0.5 (no skill) and 1.0 (perfect skill). When equal weight is given to recall and specificity, the optimal threshold is the threshold value that maximizes the Youden's $J := \text{recall} + \text{specificity} - 1$ statistic, which describes the vertical distance between the $[0,0]$-$[1,1]$-diagonal and the associated point on the ROC curve (Youden, 1950).





### 3.2.2 Application to all profile layers

The RF classifier was trained to predict two classes (stable and unstable), i.e. a binary classification based on the known failure
layer. However, our ultimate goal was to classify stability for simulated profiles where the failure layer is not known a priori.
Hence the model must be able to assess the stability of any snow layer that does not necessarily fit into either the stable or
unstable class. We therefore defined the probability that a layer is classified as unstable as:

$$P_{\text{unstable}} := \frac{1}{n_{\text{tree}}} \sum_{i=1}^{n_{\text{tree}}} \text{vote}(\text{tree}_i) \tag{8}$$

where $n_{\text{tree}}$ is the total number of trees in the forest and $\text{vote}(\text{tree}_i) \in \{0,1\}$ is the vote of the ith tree, which is either 0 (stable)
or 1 (unstable). Using $P_{\text{unstable}}$ and its overall maximum value $P_{\text{max}} := \max(P_{\text{unstable}})$, we then explored the applicability of our
RF model to complete snow profiles in four steps:

1. We applied the RF model to all profiles from the DAV data set ($N = 443$) which passed the similarity check, including
   the profiles not used for training the model and calculated the mean of $P_{\text{unstable}}$ for all profiles in each RB-LN class.

2. To explore if the overall maximum value of $P_{\text{unstable}}$ can be used to describe the stability if the weak layer is not a priori
known, we again classified the SWISS data set profiles using the $P_{\text{max}}$ values instead of the values of $P_{\text{unstable}}$ calculated
   for the manually determined weak layers.

3. We applied the RF model to *each* layer of all simulated profiles in the DAV and SWISS data sets and evaluated the
   probability of detecting the manually picked weak layers with the local maxima of $P_{\text{unstable}}$. A local maximum was
   defined as a layer whose value of $P_{\text{unstable}}$ is greater or equal than the $P_{\text{unstable}}$ values of the two layers above and the two
layers below the layer. The probability of detection (POD) was then defined as the proportion of weak layers coinciding
   with one of the three largest local maxima of $P_{\text{unstable}}$ or one of the adjacent layers within 3 cm of these local maxima.

4. We investigated if the daily maximum of $P_{\text{unstable}}$ for five winter seasons (2014-2015 and 2018-2019) at the AWS Weiss-
   fluhjoch (2536 m a.s.l.) were related to avalanche observations from the region of Davos. To this end, we compared the
   distributions of the values of $P_{\text{max}}$ on avalanche days and non-avalanche days from 1 December to 1 April of the respec-
tive winter season. Furthermore, we qualitatively compared the evolution of $P_{\text{max}}$ during the winter seasons 2016-2017
   and 2017-2018 with the avalanche activity index (AAI) for the region of Davos.

## 4 Results

### 4.1 Model development and optimization

Using the complete set of features and default hyperparameters resulted in a five-fold cross-validated accuracy of $86 \pm 6\%$ for
the classification of unstable and stable profiles in the DAV training data set ($N = 146$, balanced). Removing highly correlated
features (Pearson's r > 0.8), and conducting a first round of hyperparameter optimization, the mean accuracy increased to



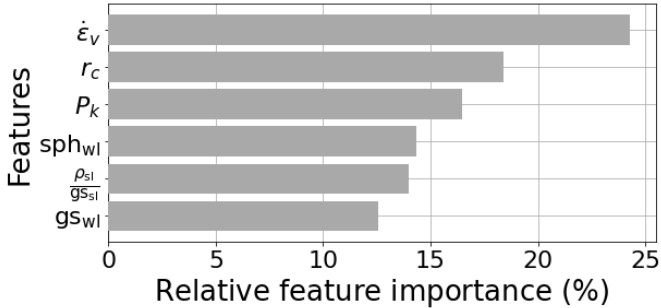

**Figure 6.** Feature importance ranking for the final model, based on evaluating the Gini impurity decrease at each split for every tree in the RF. Most important features were: Viscous deformation rate ($\dot{\epsilon}_v$), critical cut length ($r_c$), skier penetration depth ($P_k$), sphericity of grains in the weak layer (sph$_{wl}$), ratio of mean slab density and grain size ($\frac{\rho_{sl}}{gs_{sl}}$) and weak layer grain size (gs$_{wl}$). For further details on these features, see Appendix B (Table B1).

$88 \pm 8\%$. The feature importance ranking obtained with this first optimized model is shown in Appendix B (Figure B1). To enhance the interpretability of the model, we removed all features with relative feature importance lower than 5 %, resulting in six features. A further optimization of hyperparameters then yielded a model with a five-fold cross-validated accuracy of

$88 \pm 6\%$. We used these optimized hyperparameters and the reduced number of features to train the final model on the complete set of unstable and stable profiles in the DAV data set. The feature importance ranking for the six features of the final model are shown in Figure 6 and the final hyperparameters are presented in Appendix B (Table B2).

## 4.2    Model evaluation

### 4.2.1    Performance assessment with the SWISS data set

We evaluated the performance of the RF model by classifying the manually defined weak layers for the profiles from the SWISS data set. Figure 7b displays a contingency table with the predicted labels using the default classification threshold of 0.5 and Table 1 shows the resulting performance measures. The overall accuracy was $88\%$, 68 of the 75 unstable weak layers were correctly classified (recall of $91\%$), and 39 of 46 stable weak layers were classified correctly (specificity of $85\%$). The precision value was high ($91\%$), as only 7 of the 75 profiles predicted as unstable were stable according to the ground

truth label. Although the classification threshold of 0.5 resulted in good model performance, the optimal threshold value maximizing the Youden's J statistic was 0.71 (compare orange and red dots in Fig. 7a). With a threshold of 0.71, precision and specificity scores improved at the expense of the recall value (Table 1). From an operational perspective, it is thus questionable whether the increased number of false negative predictions associated with this Youden index optimization indeed represents an improvement.





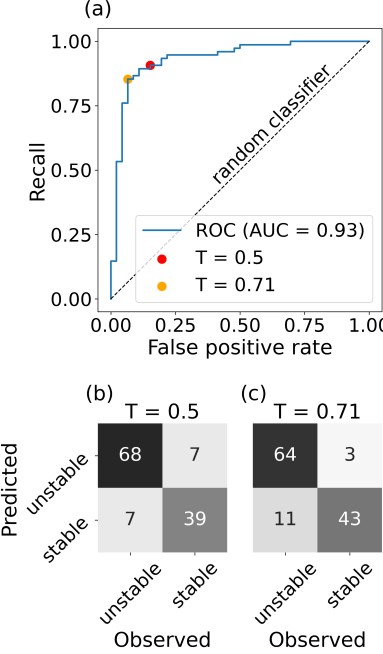

**Figure 7.** (a) ROC curve analysis and (b-c) contingency tables for the classification of the manually determined weak layers of the SWISS data set. Contingency tables are shown for (b) the default threshold ($T = 0.5$) and (c) the optimized threshold ($T = 0.71$) obtained from the ROC curve analysis.

**Table 1.** Performance measures for the classification of profiles from the SWISS data set based on the manually determined weak layers and using two different thresholds ($T$): 0.5 (default) and 0.71 (optimized).

| Performance measure | $T = 0.5$ | $T = 0.71$ |
|---|---|---|
| accuracy | 88 % | 88 % |
| precision | 91 % | 96 % |
| recall | 91 % | 85 % |
| specificity | 85 % | 93 % |

### 4.2.2 RF model applied to other stability classes

We determined $P_{\text{unstable}}$ (Eq. 8) for all manually selected weak layers from the DAV data set, and computed mean values for each RB-LN subgroup (Figure 8a). From the unstable training class in the upper left of the RB-LN diagram, to the stable training class in the lower right, the mean values of $P_{\text{unstable}}$ form an inclined plane over all other classes, which were not considered in the training of the RF model. Values decrease from top to bottom, i.e. from higher to lower LN values, and from




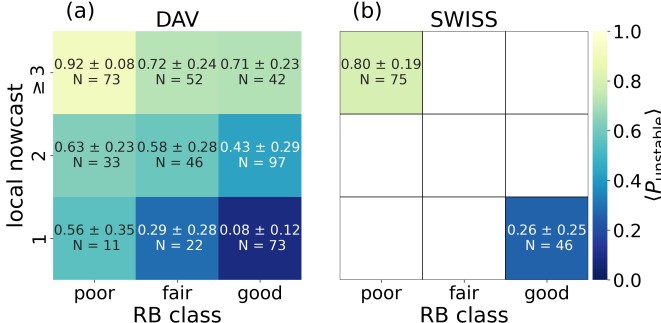

**Figure 8.** Average values $\langle P_{\text{unstable}} \rangle \pm \sigma$ of the probability of instability over all manually determined weak layers shown for each RB-LN class of (a) the DAV data set and (b) the SWISS data set.

the left to the right, i.e. from higher to lower RB stability. Considering only the RB stability classes of both data sets not used for training, $P_{\text{unstable}}$ decreased from poor stability (mean: 0.73, 67% of 119 profiles [DAV: 44, SWISS: 75] predicted as unstable [i.e. $P_{\text{unstable}} \geq 0.71$]) to good stability (mean: 0.45, 29% of 185 profiles [DAV: 139, SWISS: 46] predicted as unstable). The decrease was even more pronounced between LN $\geq 3$ (mean: 0.76, 72% of 169 profiles [DAV: 94, SWISS: 75] predicted as unstable) compared to LN $= 1$ (mean: 0.31, 14% of 79 profiles [DAV: 33, SWISS: 46] predicted as unstable), suggesting that $P_{\text{unstable}}$ of the manually detected weak layers in the simulated profiles correlated more strongly with the local danger level estimate (LN) than with the observed stability at a point as assessed with a RB test.

Overall, these results for $\langle P_{\text{unstable}} \rangle$ suggest that our RF classifier provides valuable information on snow instability for two reasons. First, weak layers associated with lower stability in terms of the RB class had higher values of $P_{\text{unstable}}$. Second, higher values of the observed local nowcast increase the likelihood that the associated simulated profile indeed exhibits unstable properties, which was also reflected in higher $P_{\text{unstable}}$ values. Note that both the observations and simulations contain uncertainty that is difficult to quantify. This is reflected in relatively high values for the standard deviations of $P_{\text{unstable}}$, which typically were in the range of $20 - 30\%$.

### 4.2.3 RF model applied to complete snow profiles

Figure 9 shows three examples of $P_{\text{unstable}}$ calculated for all layers in various snow profiles, except the uppermost layer, which has no overlying slab layers (black line, right-hand side of subplots). These examples indicate that typical weak layers, such as depth hoar, surface hoar or soft faceted layers yield higher values of $P_{\text{unstable}}$ than layers consisting of rounded grains, melt-freeze crusts and harder layers of facets. Indeed, the mean value of $P_{\text{unstable}}$ over all layers of persistent grain types with hand hardness $\leq 2$ (4 fingers) in both data sets was $0.37 \pm 0.3$, while the average value of $P_{\text{unstable}}$ for layers consisting of rounded grains or melt-freeze crusts was $0.17 \pm 0.17$. The high standard deviation for the layers of persistent grain types suggests that the stability of such a layer is not only determined by its own properties, but also depends on the overlying slab. New snow layers (i.e. precipitation or defragmented particles) reached highest average values of $P_{\text{unstable}}$ ($0.52 \pm 0.26$). We further observed





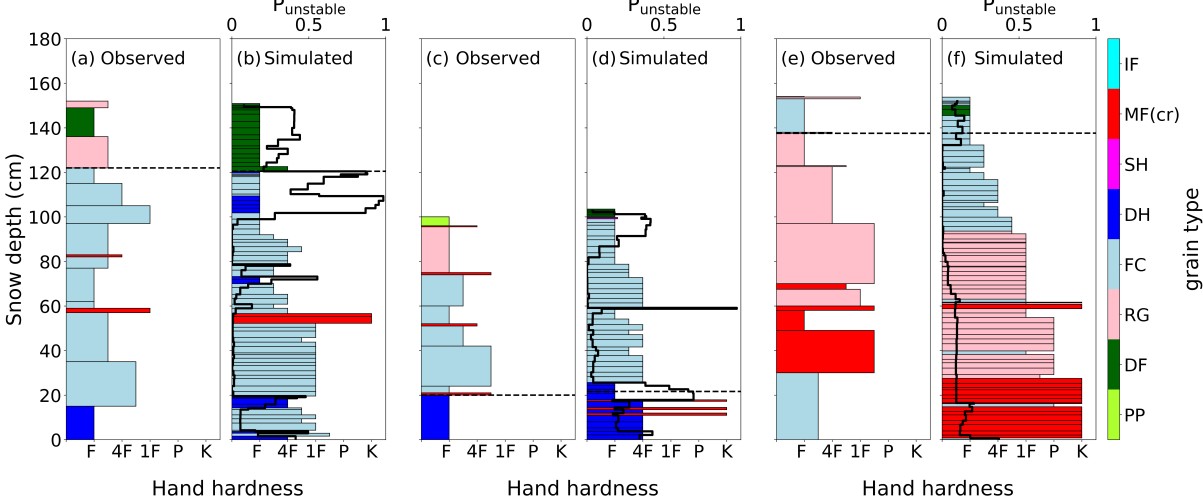

**Figure 9.** Three profile pairs from the DAV data set with (a, c, e) manually observed snow profiles and (b, d, f) corresponding simulated snow stratigraphy obtained with SNOWPACK. Hand hardness and grain type (colors) were coded after Fierz et al. (2009) (for further explanation see caption of Figure 3). The dashed horizontal line in the manual profile displays the observed height of the RB failure layer and the corresponding manually determined layer is indicated by a dashed line in the respective simulated profile. The black line in the simulated profiles shows the probability of instability $P_{\text{unstable}}$ determined for each layer. Observed RB results were (a) RB score 2, whole block, (b) RB score 6, edge and (c) RB score 4, partial. The local danger level estimates were (a) LN = 3, (b) LN = 2 and (c) LN = 1.

simulated layers with high values of $P_{\text{unstable}}$, i.e. potential weak layers, which were not observed in the manual counterpart (e.g. Figure 9c/d: surface hoar present in simulated, but not in manual profile).

To explore if the overall maximum value of $P_{\text{unstable}}$ can be used to describe the stability when the weak layer is not a priori

known, we determined $P_{\text{max}} := \max(P_{\text{unstable}})$ for each profile of the SWISS data set. Using $P_{\text{max}}$ and a default threshold of 0.5, we classified the profiles as unstable and stable. The resulting contingency table is shown in Figure 10b and the associated performance measures are shown in Table 2. With this threshold value, the classifier performed well in labeling unstable profiles as unstable (recall = 96%), but almost half of the stable profiles were misclassified (specificity = 55%). The optimal threshold value for $P_{\text{max}}$ was 0.77 (orange dot in Fig. 10c), greatly improving the overall performance (third column in Table

2, all performance measures > 90%). This optimal threshold value of 0.77 was close to the optimized value obtained for the classification of the manually selected weak layers (i.e. 0.71, Sect. 4.2.1) which led to similar values of the performance measures (second column in Table 2).

### 4.2.4 Weak layer detection

To investigate if our RF model can be used to detect the weakest layer within a profile, we calculated the probability of

detecting the manually picked weak layers with the local maxima of $P_{\text{unstable}}$ as described in Sect. 3.2.2 (point 3). For the DAV data set, the overall POD was 60%, and POD values strongly varied between different RB-LN classes (Figure 11a). While for





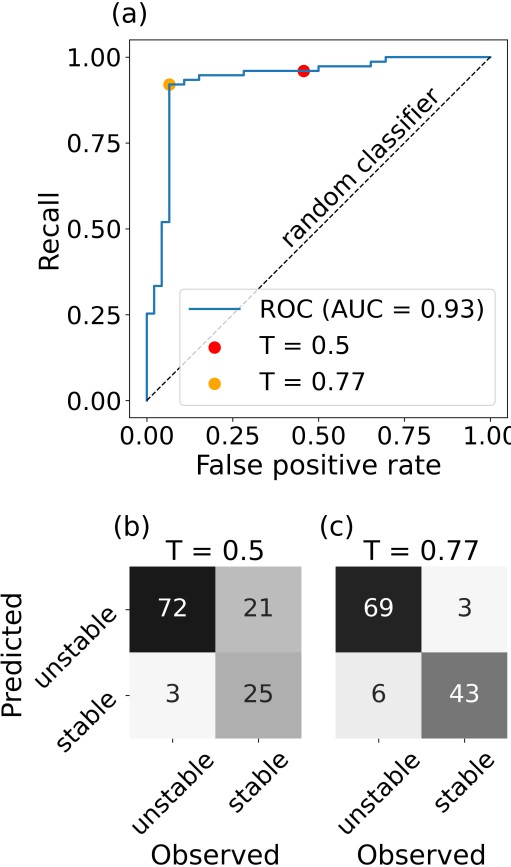

**Figure 10.** (a) ROC curve analysis and (b-c) contingency tables for the classification of the unstable and stable classes from the SWISS data set, using the maximum value of the probability of instability, $P_{max}$. Contingency tables are shown for (b) the default threshold ($T = 0.5$, red dot in a) and (c) the optimized threshold ($T = 0.77$, orange dot in a) obtained from the ROC curve analysis.

**Table 2.** Performance measures for the classification of profiles from the SWISS data set based on the maximum value of the probability of instability ($P_{max}$) for different classification thresholds ($T$): 0.5 (default), 0.71 (optimized value for the classification of manually selected weak layers) and 0.77 (optimized value for the classification based on $P_{max}$).

| Performance measure | $T = 0.5$ | $T = 0.71$ | $T = 0.77$ |
|---|---|---|---|
| accuracy | 80 % | 92 % | 93 % |
| precision | 77 % | 93 % | 96 % |
| recall | 96 % | 93 % | 92 % |
| specificity | 55 % | 89 % | 93 % |

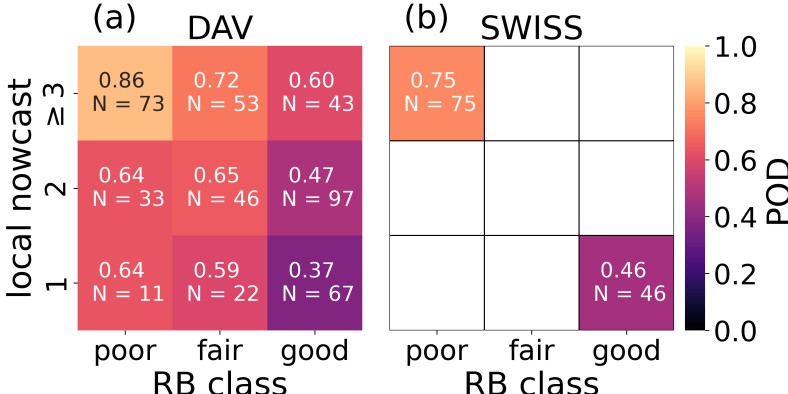

**Figure 11.** Probability of detecting the weakest layer with the three largest local maxima of $P_{\text{unstable}}$ and adjacent layers within 3 cm of these local maxima shown for every RB-LN class of (a) the DAV and (b) the SWISS data set.

the unstable training class, the POD was high ($86\%$), the POD was low ($37\%$) for the stable training class. For the SWISS data set, the POD was $75\%$ for the unstable class and $46\%$ in the stable class (Figure 11b). Lower POD values for the classes with higher stability can be explained by the fact that the manual identification of the simulated layer associated with the RB failure layer was generally less clear, since a prominent weak layer was not present. In addition, «weak» layers which only failed with a large additional load (high RB score) and which result in a fracture not propagating (a partial RB failure), are usually not associated with instability (Schweizer and Jamieson, 2003). Thus, it seems plausible that distinguishing these (not truly) weak layers from other layers within a profile is more difficult; yet it is also likely to be less relevant.

An important aspect regarding the weak layer detection with our RF model is the absence of any feature explicitly describing slab thickness. However, it is well known that weak layers associated with skier-triggered avalanches are typically within the first meter from the snow surface (e.g. Schweizer and Camponovo, 2001; van Herwijnen and Jamieson, 2007). To account for this, we investigated if adding information on slab thickness improved the weak layer detection. To this end, we defined the function

$$P^*_{\textbf{unstable}}(w) = P_{\textbf{unstable}}[(1-w) + w \cdot \frac{\text{pde}(D_{\text{slab}})}{\text{pde}_{\text{max}}}] \tag{9}$$

which includes a weighting factor $w$ and the normalized estimated probability density function $\text{pde}(D_{\text{slab}})$ of the observed slab thicknesses $D_{\text{slab}}$ in the DAV data set (Figure 12a). We analyzed the influence of the weighting factor $w$ on the probability of detecting the manually determined weak layer with the maximum value $P^*_{\text{max}}(w) := \max(P^*_{\text{unstable}}(w))$. We counted a weak layer as detected, when $P^*_{\text{max}}(w)$ was located within 3 cm of the manually picked weak layer. To calculate the POD, we only considered the unstable classes of the DAV and SWISS data set. For $w = 0$, i.e. when not accounting for slab thickness, the POD was $55\%$ for the DAV and $44\%$ for the SWISS data set. The largest POD values of $67\%$ (DAV data set) and $57\%$ (SWISS data set) were achieved for weights of $w = 0.14$ and $w = 0.12$, respectively. For larger weighting factors, the POD decreased again (Figure 12b). Thus, accounting for slab thickness increased the probability to detect a weak layer found with a RB test,





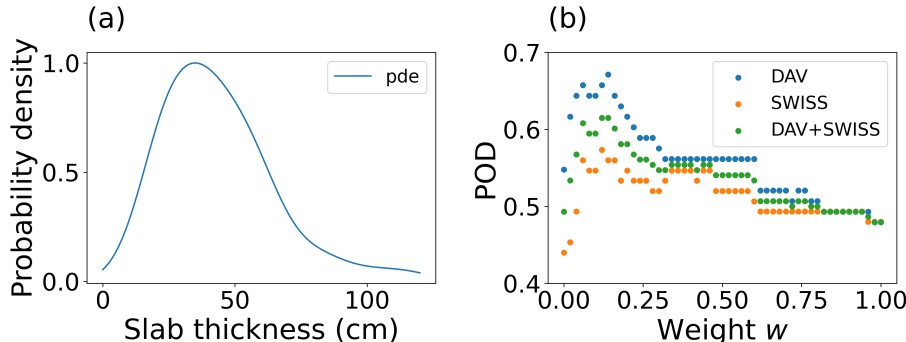

**Figure 12.** (a) Normalized Gaussian kernel density estimate (pde) for the distribution of observed slab thicknesses in the unstable class of the DAV data set. (b) Probability of detecting the manually picked weak layer in dependence of the weighting factor $w$ used in the calculation of $P^*_{\mathrm{unstable}}(w)$ (eq. 9) for the unstable classes in the DAV data set (blue markers), in the SWISS data set (orange markers) and in the combination of both (green markers).

and hence a weak layer which can potentially be triggered by a human. On the other hand, the relatively low values of $w$ for the highest POD values suggest that with our model, accounting for slab thickness is of only limited importance.

### 4.2.5 Comparison with avalanche activity

To demonstrate the practical applicability, we applied the RF model to SNOWPACK simulations for five winter seasons (2014-2015 to 2018-2019) driven with meteorological data from the AWS at the Weissfluhjoch study site at 2540 m a.s.l. For these five winter seasons (597 days), values of $P_{\mathrm{max}}$ were significantly higher on avalanche days (median $= 0.88$) than on non-avalanche days (median $= 0.51$; Mann–Whitney U test, $p < 0.001$; Figure 13). Applying the threshold value of 0.77 to the daily values of $P_{\mathrm{max}}$ yielded an overall accuracy of 73% for the discrimination between avalanche days and non-avalanche days. Of the 252 avalanche days, 69% occurred on days exceeding the threshold, while for 75% of the 345 non-avalanche days $P_{\mathrm{max}}$ was below the threshold.

Two examples for the temporal evolution of the simulated snow stratigraphy in terms of grain types, values of $P_{\mathrm{unstable}}$ and $P_{\mathrm{max}}$ over entire winter seasons at the WFJ are shown in Figures 14 (winter 2016-2017) and 15 (winter 2017-2018) in comparison to the avalanche activity index AAI of observed avalanches in the region of Davos.

The 2016-2017 winter season was characterized by below average snow depth and the presence of three prominent persistent weak layers throughout the season (dark blue layers in Figure 14c). The daily maximum $P_{\mathrm{max}}$ was often located in the vicinity of these persistent weak layers (black line in Figure 14c). Three larger precipitation events in early January, early February and in mid-March were associated with increased avalanche activity (blue bars in Figure 14b). These periods of increased avalanche activity all occurred when $P_{\mathrm{max}}$ exceeded the threshold value of 0.77 (yellow shaded regions in Figure 14b).

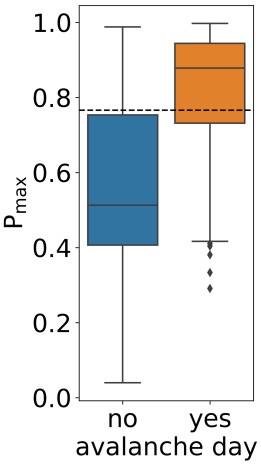

**Figure 13.** Distribution of maximal values $P_{max}$ of the probability of instability $P_{unstable}$ calculated for the simulated snow stratigraphy at the location of the AWS Weissfluhjoch (WFJ, 2540 m a.s.l.) on avalanche days and non-avalanche days during the winter seasons 2014-2015 to 2018-2019. Avalanche days were defined as days with at least one recorded dry-snow avalanche in the region of Davos, which was greater than avalanche size class one and either released naturally, was human-triggered or had an unknown trigger type. Boxes show the interquartile range from the first to third quartiles and the horizontal line displays the median. The upper and lower whiskers mark 1.5 times the interquartile range above the third and below the first quartiles, respectively. The dashed line displays the classification threshold $T = 0.77$. Number of avalanche days: $N = 252$, number of non-avalanche days: $N = 345$.

The 2017-2018 winter season was characterized by above average snow depth and a lack of persistent weak layers; $P_{max}$ was generally located below the recent new snow (black line in Figure c). Three large snowfall events between December and the middle of January resulted in three distinct avalanche periods, all of which corresponded to $P_{max}$ values exceeding the threshold value of 0.77 (yellow shaded regions in Figure b). Overall, this qualitative comparison suggests that our RF model provides valuable information linked to regional avalanche activity.


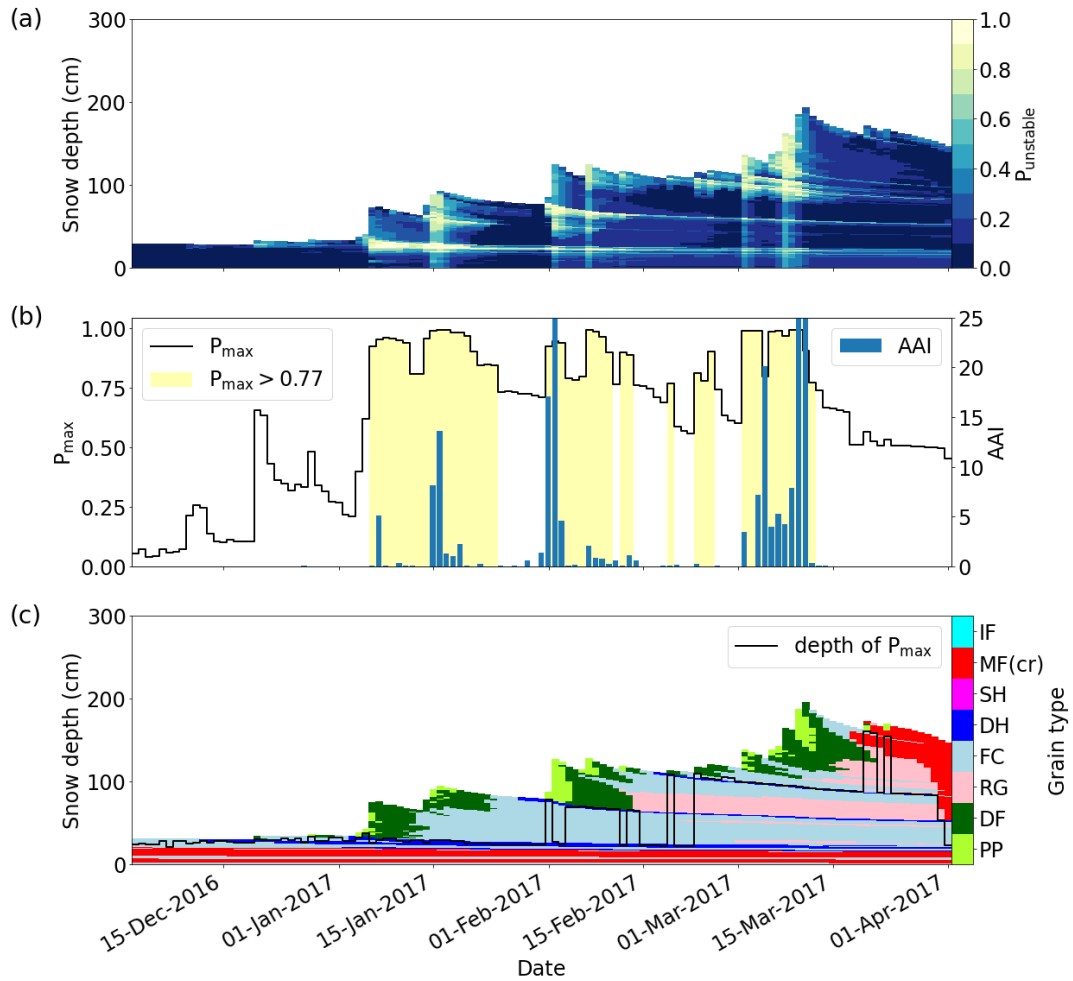

**Figure 14.** Evolution of (a) probability of instability $P_{unstable}$ (colors), (b) maximal values $P_{max}$ (black line), avalanche activity index (AAI, blue bars) and (c) the depth of $P_{max}$ (black line) and grain types (colors) calculated for the simulated snow stratigraphy at the AWS Weiss-fluhjoch (WFJ, 2540 m a.s.l.) during the winter season 2016-2017. Grain types were coded after Fierz et al. (2009) (c.f. caption of Figure 3). Yellow shaded areas in (b) indicate days with $P_{max}$ exceeding the threshold $T = 0.77$.



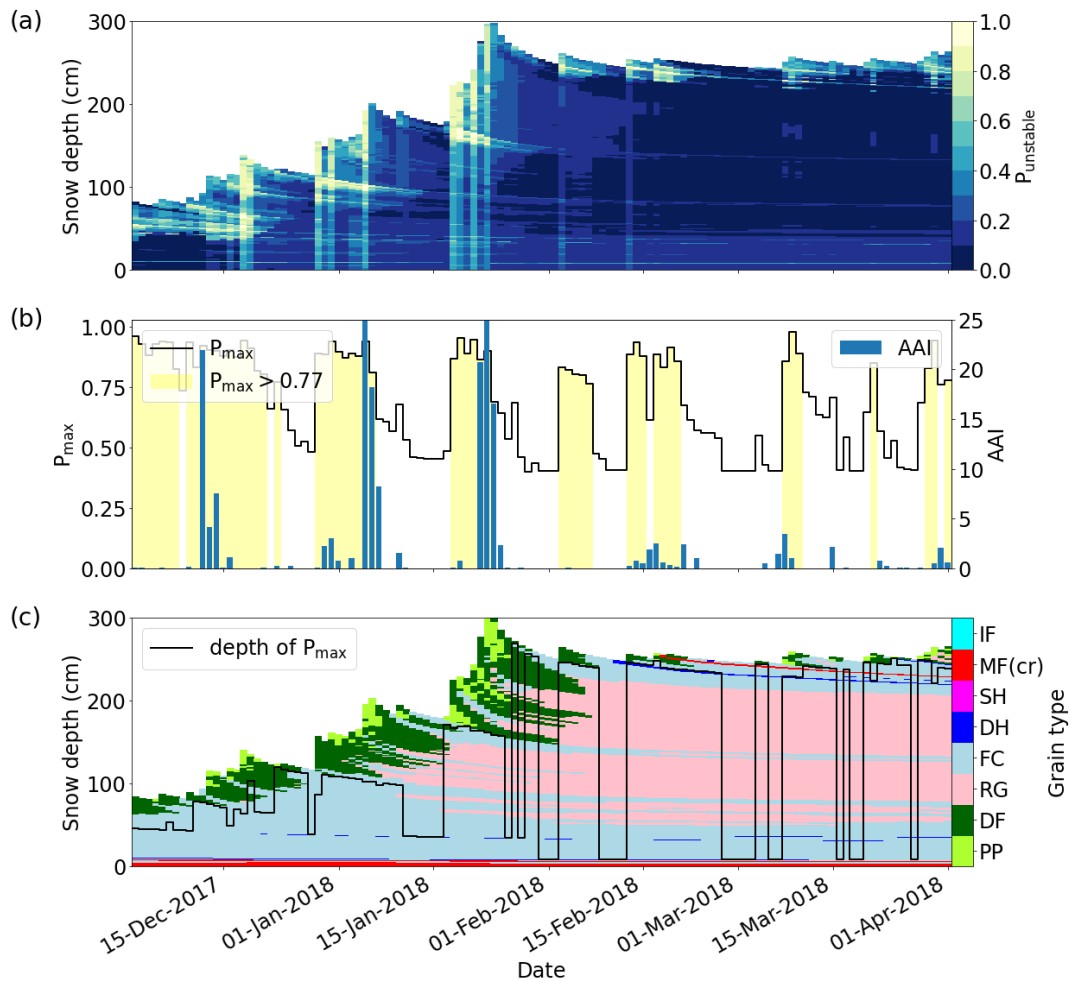

**Figure 15.** Evolution of (a) probability of instability $P_{\text{unstable}}$ (colors), (b) maximal values $P_{\text{max}}$ (black line), avalanche activity index (AAI, blue bars) and (c) the depth of $P_{\text{max}}$ (black line) and grain types (colors) calculated for the simulated snow stratigraphy at the AWS Weiss-fluhjoch (WFJ, 2540 m a.s.l.) during the winter season 2017-2018. Yellow shaded areas in (b) indicate days with $P_{\text{max}}$ exceeding the threshold $T = 0.77$.



## 5 Discussion

We trained a RF classifier to distinguish between unstable and stable snow profiles simulated with the snow cover model SNOWPACK. The resulting model provides a probability of instability for every single layer of a snow profile using six simulated features describing the layer and the overlying slab. To train and validate the model, we relied on data from manual
snow profiles with RB tests, and we compared the results from our RF model to avalanche activity in the region of Davos.

### 5.1 Data

A critical component for the construction of the RF model was a data set that allowed linking observed and modeled snow instability. We therefore established a one-to-one comparison of 742 pairs of observed snow profiles with profiles simulated at or near the location of the manual profile. These snow cover simulations were either driven with interpolated meteorological
data or with measurements from an AWS in the vicinity of the manual profile projected to virtual slopes which do not account for the influence of the surrounding terrain. Thus, the simulations cannot be expected to reproduce the exact snow stratigraphy as observed at the locations of the manual snow profiles. In particular, manual snow profiles are preferentially conducted at locations expected to exhibit poor stability (targeted sampling), e.g. slopes with below average snow depth (McClung, 2002; Techel et al., 2020a). With the scaling of the precipitation input using the ratio of observed and modeled snow depth from
pre-simulations, we intended to align modeled with observed snow depths. However, this scaling method mimics local snow redistribution on a very basic level only, and cannot replace the application of high-resolution wind fields required to explicitly simulate snow drift. While of the DAV profile pairs only 13% did not meet the predefined similarity criteria, 47% of the SWISS profiles were excluded, indicating that the interpolation of meteorological data from several stations to the exact profile location led to a better representation of the local snow stratigraphy than merely simulating the snowpack at a single nearby AWS. Our
approach of comparing profiles was based on the manual selection of a simulated layer corresponding to the observed RB failure layer and thus contained a certain degree of subjectivity. While there are automated methods for profile comparison (e.g. Hagenmuller et al., 2018; Herla et al., 2021), these were mostly developed to align complete profiles. Yet, in our study, we focused on the matching of weak layer and slab, neglecting the lower part of the profiles. Moreover, when the alignment was not obvious based on the comparison of grain type and hardness, we also considered grainsize and density to identify the weakest
layer. These additional parameters are not included in the currently available automated methods for profile comparison, and we therefore chose the manual approach.

### 5.2 Target variable

As with any classification task, the definition of a suitable target variable was crucial. In the field, instability is evaluated using a stability test, such as the RB test. We combined the observed RB test result from the manual profiles with an estimate of
avalanche danger (local nowcast) to build a binary target variable describing stability at both ends of the stability spectrum (stable vs. unstable, Tab. 2). While past studies (Gaume and Reuter, 2017; Monti et al., 2014) used only observed stability test results to train or evaluate snow instability models, exclusively relying on the observed RB test result as target variable was not



appropriate in our case. In the mentioned studies, either only observed data were considered, or the stability test was conducted
next to the AWS where the simulation was run. In our study, however, the snow cover simulations in the training data set
were driven with interpolations of meteorological data. Due to the reasons described in Sect.5.1, the simulated properties of
the snow profiles, which yielded the explanatory variables for the classification task, thus cannot fully capture the peculiarities
of the snowpack at the observation site. By considering the local nowcast assessment of avalanche danger as an additional
criterion, we selected those profiles that were likely to represent either rather stable or rather unstable conditions. As illustrated
in various studies (e.g. Techel et al., 2020b; Schweizer et al., 2021b), the proportion of poor stability test results increases with
the local danger level. Consequently, a profile with poor stability can be assumed to be more representative of the conditions
at considerable danger (level 3), and consequently to be better captured by the SNOWPACK simulation, compared to a poor
stability test result obtained at low danger (level 1).

### 5.3   Explanatory variables

We reduced the explanatory input variables of our RF model to six features while maintaining a high classification perfor-
mance. Two features combine slab and weak layer properties, namely the critical cut length, and the viscous deformation rate.
Moreover, two features are related to microstructural weak layer properties (grain size and sphericity), one feature describes
snow surface and upper slab conditions (skier penetration depth) and one feature relates to bulk slab properties (mean den-
sity divided by mean grain size). The combination of these parameters fits well with our conceptual understanding of snow
instability.
Viscous deformation rate was the most important feature in our model (Figure 6). It is proportional to the normal stress of the
slab, and inversely proportional to the viscosity (Appendix B, Table B1). High viscous deformation rates can thus occur during
loading (i.e. snowfall), and in particular in layers with low viscosity, such as layers composed of low-density new snow. In our
training data set, viscous deformation rates were significantly higher for unstable layers than for stable layers (Mann–Whitney
U test, $p < 0.001$).
In the context of human-triggered avalanches, the importance of skier penetration depth is well established (e.g. Schweizer
and Camponovo, 2001; Jamieson and Johnston, 1998). Large penetration depths increase the stress exerted on potential weak
layers deeper in the snowpack and thereby facilitate the triggering of these layers. Schirmer et al. (2010) found skier penetration
depth to be the most important variable to classify simulated snow profiles as unstable using a single classification tree model.
The parameterization of the skier penetration depth in SNOWPACK is inversely related to the mean density of the upper 30 cm
of the snow cover, and thus relates to slab properties (Schweizer et al., 2006). Changes in the penetration depth are therefore
closely linked to the presence of new snow. In our RF model, a second feature characterizing the slab was the ratio of mean
slab density to mean slab grain size. We assume that this parameter was important as it can distinguish cohesionless slabs
(low density new snow consisting of large grains) from well bonded slabs (higher density consisting of small rounded grains)
typically associated with slab avalanches.
The importance of the critical cut length $r_c$ in our RF model is in line with a recent study by Richter et al. (2019), who
observed that minimal values in modeled critical cut length of simulated profiles often coincided with observed persistent





weak layers. As such, it is likely that the critical cut length in our model favors the classification of persistent weak layers as unstable. While the critical cut length is related to crack propagation, our set of features did not include any parameter related to failure initiation. Indeed, the traditional skier stability index $SK_{38}$ (Föhn, 1987b; Jamieson and Johnston, 1998;

Monti et al., 2016) and the related failure initiation criterion (Reuter et al., 2015a) appeared at the lower ranks of the feature importance ranking (Appendix B, Figure B1). Recently, Reuter et al. (2022) suggested using a combination of the critical cut length and a failure initiation index to differentiate stable from unstable profiles using a threshold-based approach. Applying these thresholds to our DAV data set resulted in a recall of $42\%$, much lower than our RF model (Table 1). While this suggests that combining failure initiation and crack propagation indices in SNOWPACK has low predictive power, we cannot exclude

that these results are biased by uncertainties introduced by the manual identification of the weak layers in the SNOWPACK simulations.

### 5.4   Training and evaluating the RF model

For the construction of the RF model, we used the DAV data set, which included both detailed observations of the snow cover and its stability (Schweizer et al., 2021b), as well as high-quality meteorological input for the SNOWPACK simulations from

a dense network of AWS. However, the number of profile pairs in the stable and unstable classes was rather small ($N = 67$ and $N = 73$, respectively). Due to this limited amount of data points, we conducted feature selection, hyperparameter optimization and training of the RF model all on the same data set without further splitting. This resulted in a five-fold cross-validated accuracy of 88 % on the balanced DAV training data. Schirmer et al. (2010) achieved a cross-validated accuracy of 75 % when training a classification tree to distinguish between rather stable and rather unstable simulated profiles. However, their

definition of the target variable differed from ours and their data set was imbalanced.

The validation of our model on a second independent data set (SWISS) revealed a robust performance (overall accuracy: $88\%$) in the binary classification of the manually determined weak layers. Optimal performance with respect to the Youden's J statistic was reached with a classification threshold of 0.71, the default threshold (0.5) used in the training configuration, however, led to the same overall accuracy. For any application of the model, the threshold should hence be adjusted according

to the specific requirements on detection and false alarm rate. To overcome the subjectivity inherent in the manual identification of weak layers in the simulated profiles, we again classified the SWISS profiles using the maximum value of $P_{unstable}$ among all layers of the profile. With an optimized classification threshold (0.77), this classification yielded an accuracy of $93\%$. This approach, using the maximum value of $P_{unstable}$, thus led to a better classification performance than using the manually selected weak layers. The high optimal threshold value of 0.77 could be due to the fact that some weaker layers in the simulations

were not present in the manual profiles. Furthermore, this shift in threshold values might also be related to differences between training and validation data set: The training profiles were all located in the region of Davos, an area characterized by an inner-alpine snow climate (e.g. Schweizer et al., 2021b). While for 64 % of the manual profiles in the training data set, the RB failure interface was adjacent to a layer including persistent grain types, this was the case for only 45 % of the profiles in the validation data set, which were conducted in various snow climatological regions within Switzerland.



## 5.5 Model strengths and limitations

Applying our RF model to snow layers not falling into the stability categories of the binary target variable produced reasonable results (Figures 8, 9). Moreover, the detection of weak layers performed well under poor stability conditions (Figure 11). While previous studies (Schweizer et al., 2006; Schirmer et al., 2010) used separate routines for weak layer detection and instability assessment, our approach offers the possibility of assessing instability and detecting the weakest layer with one single index, the maximum of the probability of instability over all layers of the simulated snow profile.

Clearly, the interpretability of our RF model is constrained by its black-box character. However, an advantage of RF models is the ability to capture complex multi-variable relationships between features and target variable, beyond linear or threshold-based dependencies. Moreover, our model is built on only six features, which facilitates its application. An apparent limitation of our method is the lack of profiles with intermediate stability in the training data, which prevents a direct interpretation of the absolute values of the probability of instability. The probability of instability does not directly refer to a physical quantity, but should always be interpreted as a mean vote of trees which were trained with profiles from both ends of the stability spectrum. Setting thresholds to differentiate fair from poor or good stability would require more training data. Nevertheless, the comparison of modeled snow instability with observed avalanche activity for entire winter seasons at the WFJ revealed the potential of our model to indicate conditions of poor stability by using the optimized threshold value from the binary classification. The transferability of our RF model and its optimized threshold to other snow climatological settings should be evaluated on further independent data sets.

## 6 Conclusion and outlook

We introduced a novel method to assess dry-snow instability from simulated snow stratigraphy. Our Random Forest (RF) model provides a probability of instability $P_{\text{unstable}}$ for each layer of a snow profile simulated with SNOWPACK, given six input variables describing microstructural, macroscopic and mechanical properties of the particular layer and the overlying slab. The probability of instability allows detecting the weakest layer of a snow profile and assessing its degree of instability with one single index, a main advantage of this new model. Although the RF model was trained with only 146 layers manually labeled as either unstable or stable, it classified profiles from an independent validation data set with high reliability (accuracy: 88%, precision: 96%, recall: 85%) using manually predefined weak layers and an optimized classification threshold. The binary classification performance with optimized threshold was even higher (accuracy: 93%, precision: 96%, recall: 92%), when the weakest layers of the profiles were not known a priori and were instead identified with the maximum of $P_{\text{unstable}}$. Finally, we illustrated the potential of our model and its optimized threshold value to indicate conditions of poor stability by comparing the temporal evolution of modeled snow instability with observed avalanche activity in the region of Davos for five winter seasons.

In principal, our model provides an estimate of dry-snow instability for any simulated snow profile for which the required input variables are available. For the derivation of further threshold values which detect intermediate stability, more data are required. The threshold that distinguishes rather unstable from rather stable profiles may need to be adjusted if the simulated





stratigraphy originates from models other than SNOWPACK, or if applied in a region with a snow climate strongly differing from the conditions in the Swiss Alps.

In the future, the RF model may be used to estimate avalanche danger from simulated snow stratigraphy. To this end, the
540 RF model would be applied to modeled snow stratigraphy at different locations within one region. The respective maxima of $P_{\text{unstable}}$ and the corresponding frequency distribution may then yield information on the snowpack stability as well as the spatial distribution of stability, and the depths of the weakest layers determined with these maxima may provide an indicator of the expected avalanche size. Since this application of the RF model covers all three factors contributing to avalanche hazard (Techel et al., 2020a), it could be of great value for operational avalanche forecasting. This application may even be extended by
545 extracting the grain type of the weak layer to distinguish between the avalanche problem types "persistent weak layer problem" and "new snow problem" (EAWS, 2021). Besides this operational usage, the method described is also suited for analyzing past and future changes in snow instability due to climate warming.





## Appendix A: Data

The locations of snow profiles and AWS are shown in Figures A1 and A2.

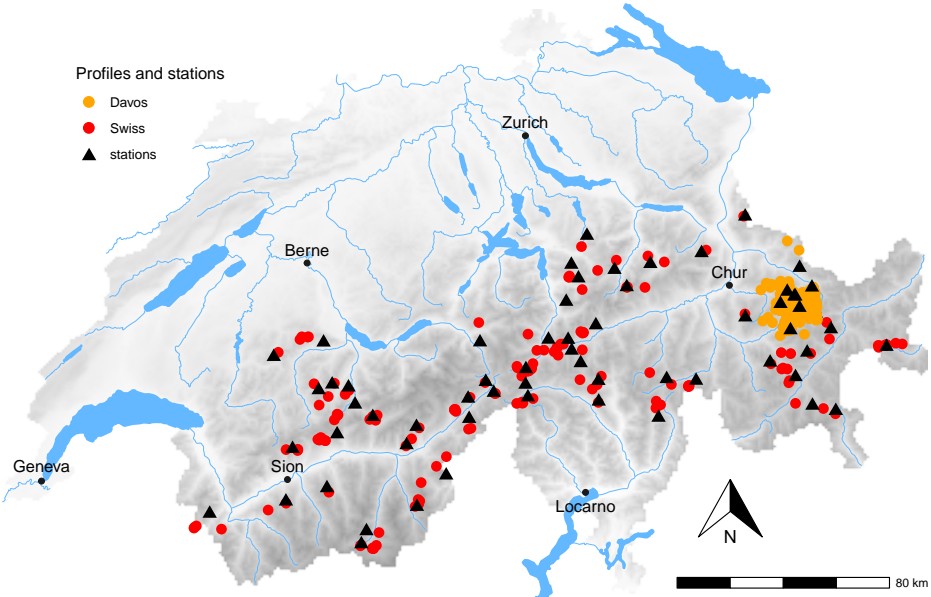

**Figure A1.** Map of Switzerland showing the locations of the snow profiles and the automatic weather stations used in the Davos and Swiss data set (orange and red markers respectively). A zoom into the Davos region is shown in Fig. A2.



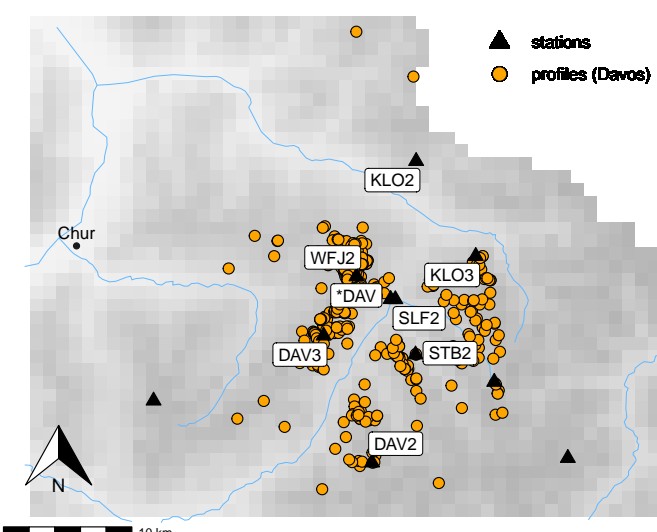

**Figure A2.** Map showing the region of Davos with the automatic weather stations (with their labels) and the profile locations (orange markers).





## Appendix B: Classification model

To build the classification model, we used 34 features which are described in Table B1. The relative importance of a subset of 20 of these features are shown in Figure B1. The final values of the hyperparameters in the RF model are compiled in Table B2.





**Table B1.** Table with all features describing slab (sl) and weak layer (wl) properties

| Abbreviation | Feature | Formula / remarks | Reference |
|---|---|---|---|
| **basic SNOWPACK output parameters** | | | |
| $gs_{wl}$ | grain size of wl | - | Lehning et al. (2002b) |
| $sph_{wl}$ | sphericity of wl | - | Lehning et al. (2002b) |
| $bs_{wl}$ | bondsize of wl | - | Lehning et al. (2002b) |
| $d_{wl}$ | dendricity of wl | - | Lehning et al. (2002b) |
| $gt_{wl}$ | grain type of wl | - | Lehning et al. (2002b) |
| $\rho_{wl}$ | density of wl | - | Bartelt and Lehning (2002) |
| $\eta$ | viscosity of wl | - | Lehning et al. (2002b) |
| $age_{wl}$ | age of wl | - | - |
| HS | snow depth | - | - |
| **composed features weak layer** | | | |
| $\frac{\rho_{wl}}{gs_{wl}}$ | - | - | - |
| $\frac{\rho_{wl} \cdot bs_{wl}}{gs_{wl}}$ | - | - | - |
| **composed features slab** | | | |
| $D_{sl}$ | slab thickness | - | - |
| $\rho_{sl}$ | mean sl density | - | - |
| $\frac{\rho_{sl}}{gs_{sl}}$ | - | with $gs_{sl}$ = mean sl grain size | - |
| $\frac{\rho_{sl} \cdot bs_{sl}}{gs_{sl}}$ | - | with $bs_{sl}$ = mean sl bond size | - |
| $\rho_{sl20}$ | mean density of 20 cm above wl | - | - |
| $\rho_{10max}$ | maximal mean density of all 10 cm windows above wl | - | - |
| $P_k$ | skier penetration depth | $P_k = 34.6/\rho_{30}$ with $\rho_{30}$ = mean density uppermost 30 cm | Jamieson and Johnston (1998), Schweizer et al. (2006) |





| Abbreviation | Feature | Formula / remarks | Reference |
|---|---|---|---|
| composed features weak layer & slab | | | |
| $\triangle$gs | difference in grain size between wl and layer above wl | - | Schweizer and Jamieson (2007) |
| $\triangle$h | difference in hardness between wl and layer above wl | - | Schweizer and Jamieson (2007) |
| $\left[\frac{\rho}{\text{gs}}\right]_{\text{wl}/(\text{wl}+1)}$ | - | $\left[\frac{\rho}{\text{gs}}\right]_{\text{wl}/(\text{wl}+1)} = \frac{\rho_{\text{wl}}\text{gs}_{\text{wl}+1}}{\text{gs}_{\text{wl}}\rho_{\text{wl}+1}}$ with $(\text{wl}+1)$: layer above wl | - |
| rts | relative threshold sum | - | Monti et al. (2014) |
| snow mechanical features | | | |
| $\tau_p$ | shear strength of wl | - | Jamieson and Johnston (1998) |
| $\sigma_n$ | normal stress exerted on wl by sl | - | Bartelt and Lehning (2002) |
| $\triangle\tau$ | skier shear stress on wl | calculated for slope angle = 38° | Jamieson and Johnston (1998) |
| $\triangle\tau^*$ | refined skier shear stress on wl | calculated for slope angle = 38° | Monti et al. (2016) |
| $\text{SK}_{38}$ | skier stability index | $\text{SK}_{38} = \frac{\tau_p}{\tau_{sl38}+\triangle\tau}$, with $\tau_{sl38} = $ shear stress on wl by overlying sl | Föhn (1987b), Jamieson and Johnston (1998) |
| $\text{SK}_{38}^*$ | skier stability index, refined version | $\text{SK}_{38}^* = \frac{\tau_p}{\tau_{sl38}+\triangle\tau^*}$ | Monti et al. (2016) |
| $S_{\text{skier}}$ | failure initiation criterion | $\frac{\tau_p}{\triangle\tau}$ | Reuter et al. (2015a) |
| $r_c$ | critical cut length (flat field) | $r_c = \sqrt{\frac{2\tau_p}{\sigma_n}}\sqrt{E'D_{sl}F_{wl}}$ with $E' = $ plain strain elastic modulus of sl and $F_{wl}$ a function of $\rho_{wl} \cdot gs_{wl}$ | Richter et al. (2019) |
| $\sigma_{ns}$ | wl neck stress | - | Lehning et al. (2002b) |
| $\dot{\epsilon}_n$ | wl neck strain rate | - | Lehning et al. (2002b) |
| $\dot{\epsilon}_v$ | viscous deformation rate | $\dot{\epsilon}_v = \frac{\sigma_n}{\eta}$ | Bartelt and Lehning (2002) |
| $S_{\text{dr}}$ | deformation rate index | $S_{\text{dr}} = \frac{\sigma_c}{\sigma_{ns}}$ with $\sigma_c = $ critical neck stress | Lehning et al. (2004) |



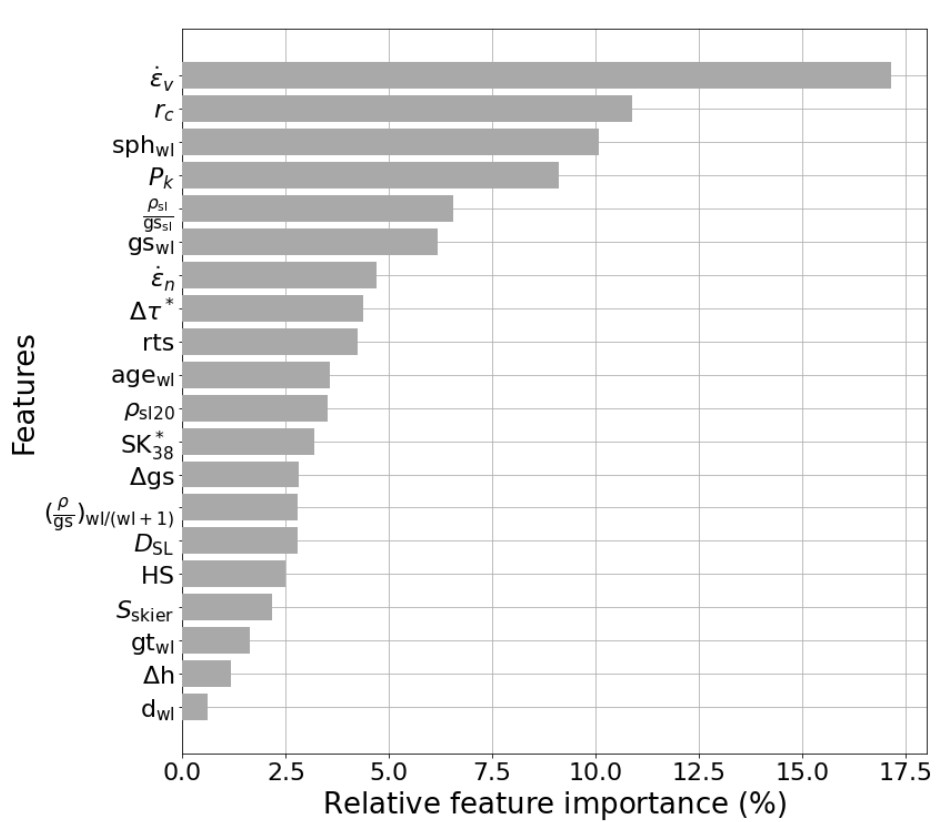

**Figure B1.** Feature importance after hyperparameter optimization round 1.





**Table B2.** Hyperparameters of final random forest model

| hyperparameter | optimized choice |
|---|---|
| Number of trees | 400 |
| Split quality measure | Gini criterion |
| Maximum depth of a tree | 7 |
| Number of features to consider at every split | $\sqrt{N_{\text{feat}}} = 6$ |
| Minimum number of samples required for a leaf node | 1 |
| Minimum number of samples required to split internal node | 3 |



*Author contributions.* AH and JS initiated this study and SM processed and analyzed the data and simulations. SM prepared the manuscript with contributions from all co-authors.

*Data availability.* The essential data sets will become available upon acceptance on the WSL data portal Envidat (www.envidat.ch).

*Competing interests.* Stephanie Mayer, Alec van Herwijnen and Frank Techel declare they have no competing interests. Jürg Schweizer is a member of the editorial board of the journal.

*Acknowledgements.* We thank Heini Wernli, Bettina Richter and Stephan Harvey for advice on model development, Florian Herla for a fruitful exchange on comparing snow profiles, Matthias Steiner and Flavia Mäder for their assistance in developing a GUI for the manual profile comparison, and Mathias Bavay for his support with the SNOWPACK model. We thank all observers and SLF staff members who contributed field observations to this study.





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
