# Peer review of "A random forest model to assess snow instability from simulated snow stratigraphy"

_The Cryosphere, 2022_

## Referee Comment (RC2)

[referee-annotated manuscript omitted]

---

## Author Comment (AC1)

**Reply to Referee #1**

The authors tackle a very important problem for the snow and avalanche community: how to provide synthetic indicators relevant for avalanche forecasting from potential huge amont of simulated snow data. I am very impressed by the obtained results. Moreover, the paper is very well written and is comprehensive with a deep analysis of the model behavior and a detailed presentation of the data pre-processing which essential for machine learning approaches. It might be sometimes difficult to catch the key results among all the results presented, but with a second read it becomes clear enough. However, the interpretation of the model explanatory variables should be qualified as the used feature importance metric is affected by correlation between the input variables and contains only partial information of the underlying « physics ». Overall, I suggest accepting this very good paper with only minor revision I have listed below.

Dear Pascal Hagenmueller

Thank you for your detailed and constructive comments on our manuscript. Please find below our replies (in blue) describing how we will address your comments in the revised manuscript.

**Minor comments:**

Abstract : add somewhere that the study domain is mainly around Davos and in Swiss.

We will add the study domain in the abstract of the revised manuscript.

L11 : give number of points in the validation data set

We will provide the number of points in the validation data set (N=121) in the revised abstract.

L14-16 : you provide the accuracy for discriminating the non avalanche / avalanche days. However, if the data is not balanced it is difficult to interpret. Use the same clear sentence as in l390-392.

We agree that only providing the accuracy is not sufficient for an imbalanced data set and will follow your suggestion when revising the abstract.

L44 : the model MEPRA (Giraud, 1992) is one of the first model that tried to combine different metrics of snow instability into a synthetic index. Add historical reference in the text.

Thank you for pointing this out. We will include this reference.

Fig. 1 : « virtual slope simulation » => « simulated snow profiles »

We will change the wording as suggested.

L83 : give reference of the rutschblock score from 1 to 7 or explain its meaning.

We will provide a reference where the test procedures are described.

L89 + 105 : « RB tests failed adjacent to layer of persistent grain types ». I do not understand what is meant here. Do you mean: the weak layer revealed by the RB test is in 64% cases composed of FC, DH or SH ?

In the data set of observed Rutschblock tests, the height where the RB failed was indicated as an interface between two observed layers. The failure layer was thus one of the two layers adjacent to this interface. In 64% of the cases in the DAV data set, one of the two layers adjacent to the failure interface was composed of persistent grain types (facets, depth hoar or surface hoar). We will clarify this in the manuscript.

L90 and throughout the text : « to evaluate the model », it is not clear what is the model here. Indeed, the « basic » model predicts whether a weak layer - slab system is unstable or not. I understand that you applied your model more extensively to simulated snow profiles. But be more specific.

We agree with you that we need to be more specific which model we are talking about. In L90 we refer to the application of the RF model on complete snow profiles. We will be more specific when using the term "model" throughout the manuscript.

Fig. 2 : x-label and plot title appear in the same form which is confusing.

We will increase the font size of the plot title to resolve this confusion.

L214 : « similarity criteria » to be defined. Do you mean criteria 1-5 ?

Yes, with similarity criteria we refer to criteria 1-5. We will add "1-5" in L214, to make this more clear.

L274-277: reword. Not clear to me. The use of the probability is not related to the fact that you want to apply the model to any layer of the profile ???

Thank you for pointing out that the wording is confusing. We will reword the sentence to clarify our intention to obtain information on layers from the complete range of the instability spectrum using the output probability that a layer is classified as unstable.

Fig. 8 : I do not understand the role of Fig.8b as the goal is here to see how the model works on intermediate instability classes.

We included Fig. 8b, which shows the average values of $P_{unstable}$ in the unstable and stable classes of the SWISS data set, as the corresponding classes of the DAV data set (in the upper left and lower right corner of Fig. 8a) were used for the training of the model. In our view, the average values of $P_{unstable}$ for the unstable and stable classes of the SWISS data set contain important additional information, as they

are independent from the training of the model and are still higher respectively lower as compared to the average values of $P_{unstable}$ from all the intermediate classes in the DAV data set.

L325-329: I did not understand your point here, could you be clearer to explain your point (L330-331).

In L325-331 we analyze mean values of $P_{unstable}$ and proportions of profiles classified as unstable for different subsets of the data not used for training: First for the two marginal RB stability classes "poor" (i.e. RB class = poor and LN $\in$ {1,2,3,4}) and "good" (i.e. RB class = good and LN $\in$ {1,2,3,4}) and then for the two marginal LN classes LN $\geq$ 3 and LN = 1, merging all RB classes (poor, fair, good). As the decrease of $<P_{unstable}>$ and the proportion of profiles classified as unstable was more pronounced from the LN $\geq$ 3 to the LN=1 subset than from the "poor" to the "good" RB class, we concluded that the simulated stability correlated more strongly with the local danger level estimate (LN) than with the observed stability at a point as assessed with a RB test. We will rewrite the paragraph in the revised manuscript to improve the clarity.

L389-390: could you plot on Fig. 13 the avalanche and non-avalanche days as defined in this paper.

We regret but do not understand what you ask us to do here.

L402: « Figure c » => « Figure 15 c » ?

Thanks for pointing out this typo. We will change to Figure 15c in the revised manuscript.

L425: « they were mostly developed to align complete profiles ». In practice, this is not true as a parameter of the model can be used to align only a sub part of the profile. In particular it is used to relax the assumption that the snow-ground interface must be matched. Besides, it is not a limit of the method since for the manually matching you also look below the weak layer for stratigraphy markers (eg. MF-crust). « these additional parameters are not included in the current available automated methods » It is implemented and shown in Viallon-Galinier et al. (2020). Actually your manual method seems to works fine enough and you do not necessarily need an automatic method. You might see the automated matching method as a further development to reduce the time spent to prepare the data but you do not need to say something wrong about the automated method limits.

Thank you for pointing out that these matching algorithms can be used to align only a sub-part of the profile and the parameters grain size and density are implemented in the model used by Viallon-Galinier et al. (2020). We will rewrite the paragraph and include this reference in the revised manuscript.

Section 5.3 : all your analysis is based on the feature importance as computed by the scipy package. First, here, you do not give any information on the « sign » (> or <) of the important feature. For instance, it is not clear (and there is no info about that) whether it is high or low values of « mean density divide by mean grain size » that promote instability. To be added. Besides, the feature importance are somehow « shared » between correlated variables. For instance, viscous deformation might be correlated to the initiation criteria such as SK38 (stress over strength) which is itself correlated to strength, stress (and so importance shared …). Your comment about the absence of

initiation criterion must therefore be qualified. Moreover, your comparison of your model score (6 parameters, training) to the « physical » model with only two parameters and no training is unfair (L. 478).

Thank you for your recommendations on how to improve Section 5.3. To include information on the "sign" of the relationship between the features and the target response, we will show partial dependence plots in the appendix of the revised manuscript. A partial dependence plot shows the effect of a given feature on the output prediction, marginalizing over the values of all other features (Friedmann, 2001).

While training the RF model we aimed at avoiding "shared" feature importance between correlated features by excluding pairs of features that were highly correlated (Pearson's r > 0.8). The correlation between viscous deformation rate and the skier stability index SK38 in our training data set was rather low (Pearson's r =- 0.19). Even when removing all features with correlation coefficients with SK38 exceeding 0.5, SK38 still appears at the lower end of the feature importance ranking.

We agree that the comparison of our trained model with the untrained threshold-based model using only the critical crack length and the initiation criterion as input features is somewhat unfair. In the revised manuscript, we will make the limitations of this comparison more explicit. We will also note that when training a decision tree of depth two on the DAV data set, the five-fold cross-validated accuracy is lower when using the critical crack length and the failure initiation criterion as compared to using only the critical crack length. This clearly indicates that for our data set the strength-over-stress initiation criteria have a very limited information content.

Fig. 13 and 14 and Section 4.2.5: the results at the regional scale are very interesting but never discussed in the paper. In particular, the model apparently failed (?) to detect clearly the big avalanche events (high AAI) at the regional scale. Add a discussion on the inherent difficulty to predict high AAI from only slab stability indices (size, spatial distribution, natural release).

We agree that there are some discrepancies between the predictions of our RF classifier and the observed regional avalanche activity and that we did not mention these results in the discussion. Our main goal with these two figures was to show the potential applicability of our RF classifier for avalanche forecasting and the overall promising results. As we only used simulations from one field site for this comparison, there can be a number of reasons why these discrepancies occur, including a lack of information on spatial snow distribution and on potential avalanche size as well as incomplete or biased avalanche data. As these are well-known problems when using avalanche observations for validation, we will briefly discuss the results shown in these figures in section 5.5, but we do not want to discuss these potential error sources in great length.

**References:**

Friedman, J.H.: Greedy function approximation: A gradient boosting machine, The Annals of Statistics, 29(5), 1189-1232, https://doi.org/10.1214/aos/1013203451, 2001.

Giraud, G.: MEPRA: an expert system for avalanche risk forecasting, Proceedings ISSW 1992. International Snow Science Workshop, Breckenridge, Colorado, U.S.A., 4-8 October 1992, pp. 97-106, 1993.

Viallon-Galinier, L., Hagenmuller, P., Lafaysse, M.: Forcing and evaluating detailed snow cover models with stratigraphy observations. Cold Regions Science and Technology 180, 103163, https://doi.org/10.1016/j.coldregions.2020.103163, 2020.

---

## Author Comment (AC2)

**Reply to Referee #2**

In "A random forest model to assess snow instability from simulated snow stratigraphy" an ensemble machine learning approach is used to classify instability in profiles from the SNOWPACK model. I enjoyed reviewing this manuscript and recommend that it be accepted subject to minor revisions based on the quality of the work and its importance to advance the field of artificial intelligence in avalanche research. I have a few thoughts for the authors to consider while preparing their final submission.

Dear Edward Bair

Thank you for your constructive comments on our manuscript. Below we describe (in blue) how we will address your comments when revising the manuscript.

1) Why are rutschblocks still being used as the test of choice? For example, Schweizer and Jamieson (2010) report unweighted average accuracies of ECTs as 0.81 - 0.95. For the rutschblock, the range is 0.67 - 0.88 when score or release type is used. Using results from a more accurate stability test might improve the performance of the random forest model used here.

We agree that the ECT is widely used among practitioners. However, for research purposes the rutschblock test is well suited and well validated data sets exist. Moreover, the rutschblock differentiates between stability classes more clearly than the ECT. As recently shown, for the very poor and poor stability classes the RB test results correlate more strongly with instability than the ECT results (Techel et al., 2020a, 2020b). For instance, Techel et al. (2020a) compared the correlation between RB and ECT and slope stability using a common data set and the same base rate of slopes rated as unstable. In contrast, the review of stability tests by Schweizer and Jamieson (2010) compared many different studies, which either explored the RB or the ECT and rarely ECT and RB in the same study. These studies used different approaches in terms of defining what stable/unstable means and how the data was selected, and hence, the base rates of unstable profiles in the data set differed. As shown by Techel et al. (2020a, Sect. 5.5) for RB and ECT, and by Brenner and Gfeller (1997) from a theoretical perspective, these definitions and the base rate have a strong impact on the resulting performance statistics. Thus, Techel et al. argued that comparisons should primarily be made when exploring stability tests using the same approach and a common data set.

2) At 27 pages with 15 figures and 2 tables, excluding the 2 appendices, the article is too long. The Cryosphere is unusually vague in article size limits, but it is expected to fit with 12 journal pages. In any case, the article's length dilutes its important findings, which show that random forests can be used to classify profiles based on stability with high accuracy. Perhaps some of the details regarding hyperparameters and explanation of the widely-used random forest model could be omitted or moved to an appendix.

We will carefully revise the manuscript, make the language more concise, and consider moving certain sections to an appendix to shorten the manuscript.

3) The finding that viscous deformation is the most important predictor is only briefly discussed. This finding deserves further discussion as it highlights how profiles alone are inadequate to classify instability.

Loading rate is one of the most important avalanche predictors, stated in Atwater and Koziol (1953) and before. The viscous deformation parameter appears to be an indirect measure of this.

Thanks for this comment. We will discuss the importance of loading rates and how these might be linked to the viscous deformation rate in more detail in the revised version.

Minor comments from the annotated PDF

L54 This is not a huge gap in the literature since SK38 and r_c have been seperetely validated. Could you provide more motivation for why evaluating both metrics together is vital?

It is correct that both indices have been validated separately. However, to predict snow instability information on both failure initiation (SK38) and crack propagation (rc), and combined threshold values are required (e.g. Reuter et al. 2015). Such an approach has not yet been investigated for simulated snow profiles.

L62 citation?

We will move the citation for the RF classification (Breiman, 2001a) from L220 to L62.

L63 I suggest deleting both instances of "rather"

We prefer to keep the terms "rather", as it is not possible to unambiguously define what stable and unstable snowpack conditions are. Obviously, "unstable" profiles were observed on slopes that did not avalanche.

L82-84 Hasn't the science moved past rutschblocks? Why are they still being used over ECTs ? For example, Schweizer and Jaimieson (2010) report unweighted average accuracies of ECTs as 0.81 - 0.95. For the rutschblock, the range is 0.67 - 0.88, when score or release type is used.

Please see answer to question 1).

L201-203 Or maybe that's because the whole concept of a weak layer that always be pointed to as the culprit in an avalanche is too simplistic. It's great to have an identifiable weak layer for studies like this, but sometimes (for example in storm slab avalanches) there is not an easily definable weak layer.

We agree with you that identifying the weakest layer can be challenging, however, we adhere to the fact that the existence of a weak layer is a prerequisite for the formation of a slab avalanche (Schweizer et al., 2003). While it is clear that for storm snow instabilities it can sometimes be difficult to identify a weak layer in a manual snow profile, this does not mean that there is no weak layer. The fact that we cannot easily identify one in our snow pits just highlights the inherent difficulties in obtaining good data from such manual measurements. However, as in our dataset most weak layers were persistent weak layers, this is not a big issue.

Moreover, in our study, we aimed at identifying the simulated layer most similar to the observed rutschblock failure layer, rather than identifying a weak layer from scratch. In L201-203 we describe that this matching was not always unambiguous, but this was rather due to differences between simulated and observed snow profiles.

L220 put this citation at the first mention of RF on l 62. Since RF is already defined there, "Random Forest" need not be spelled out here.

We will follow your suggestion in the revised manuscript.

L222 delete accounting

We will replace the wording by: ..., this model can account for complex mutual dependencies ...

L412 linked

We will replace "allowed linking" with "linked" in the revised manuscript.

L526 detection of

We will replace "detecting" with "the detection of" in the revised manuscript.

**References**

Brenner, H. and Gefeller, O.: Variations of sensitivity, specificity, likelihood ratios and predictive values with disease prevalence, Stat. Med., 16, 981–991, https://doi.org/10.1002/(SICI)1097-0258(19970515)16:9<981::AID-SIM510>3.0.CO;2-N, 1997

Schweizer, J., Jamieson, J. B., and Schneebeli, M.: Snow avalanche formation, Rev. Geophys., 41, 1016, https://doi.org/10.1029/2002RG000123, 2003.

Schweizer, J. and Jamieson, B.: Snowpack tests for assessing snow-slope stability, Ann. Glaciol., 51, 187–194, https://doi.org/10.3189/172756410791386652, 2010.

Techel, F., Winkler, K., Walcher, M., van Herwijnen, A., and Schweizer, J.: On snow stability interpretation of extended column test results, Nat. Hazards Earth Syst. Sci., 20, 1941–1953, https://doi.org/10.5194/nhess-20-1941-2020, 2020a.

Techel, F., Müller, K., and Schweizer, J.: On the importance of snowpack stability, the frequency distribution of snowpack stability, and avalanche size in assessing the avalanche danger level, The Cryosphere, 14, 3503–3521, https://doi.org/10.5194/tc-14-3503-2020, 2020b.

---

## Author Response (AR1)

We thank Guillaume Chambon, Pascal Hagenmuller, Edward Bair and an anonymous referee for their positive and constructive feedback, which we have addressed in the revised manuscript. Please find below our detailed replies referring to the changes we implemented in the revised version of the manuscript. Here, we only list those remarks that required changes in the manuscript or that have not been answered in detail in the first reply. In addition, we provide a marked-up manuscript version showing the changes made.

We hope that the manuscript is now suitable for publication in The Cryosphere.

Stephanie Mayer, on behalf of all co-authors

**Reply to Reviewer #1: Pascal Hagenmuller**

Abstract : add somewhere that the study domain is mainly around Davos and in Swiss.

We added the study domain (Swiss Alps) in the abstract of the revised manuscript.

L11 : give number of points in the validation data set

We provided the number of points in the validation data set (N=121) in the revised abstract.

L14-16 : you provide the accuracy for discriminating the non avalanche / avalanche days. However, if the data is not balanced it is difficult to interpret. Use the same clear sentence as in l390-392.

Changed as suggested.

L44 : the model MEPRA (Giraud, 1992) is one of the first model that tried to combine different metrics of snow instability into a synthetic index. Add historical reference in the text.

We included this reference.

Fig. 1 : « virtual slope simulation » => « simulated snow profiles »

We changed the wording as suggested.

L83 : give reference of the rutschblock score from 1 to 7 or explain its meaning.

We provided a reference where the test procedures are described.

L89 + 105 : « RB tests failed adjacent to layer of persistent grain types ». I do not understand what is meant here. Do you mean: the weak layer revealed by the RB test is in 64% cases composed of FC, DH or SH ?

In the data set of observed Rutschblock tests, the height where the RB failed was indicated as an interface between two observed layers. The failure layer was thus one of the two layers adjacent to this interface. We clarified this in the revised manuscript. As the DAV data set contains information on which of these two layers adjacent to the interface was the actual failure layer, we now provide the ratio of failure layers which contained persistent grain types in section 2.1.1. In the discussion section 5.4, we only provide the ratios with respect to the adjacent layers for the training subset of DAV and the validation subset of SWISS, since for the SWISS data set we only have information on the failure interface. The numbers in the revised manuscript differ from the previous ones, since we previously failed to include rounded facets in the persistent grain types.

L90 and throughout the text : « to evaluate the model », it is not clear what is the model here. Indeed, the « basic » model predicts whether a weak layer - slab system is unstable or not. I understand that you applied your model more extensively to simulated snow profiles. But be more specific.

We reviewed the usage of the term "model" throughout the manuscript and changed the wording when a more specific expression was necessary to avoid confusion.

Fig. 2 : x-label and plot title appear in the same form which is confusing.

We deleted the plot title as the information is contained in the caption.

L214 : « similarity criteria » to be defined. Do you mean criteria 1-5 ?

Yes, with similarity criteria we refer to criteria 1-5. We added "1-5", to make this clearer.

L274-277: reword. Not clear to me. The use of the probability is not related to the fact that you want to apply the model to any layer of the profile ???

We reworded the sentence to clarify our intention.

L325-329: I did not understand your point here, could you be clearer to explain your point (L330-331).

In these lines we analyzed mean values of $P_{unstable}$ and proportions of profiles classified as unstable for different subsets of the data not used for training: First for the two marginal RB stability classes "poor" (i.e. RB class = poor and LN $\in \{1,2,3,4\}$) and "good" (i.e. RB class = good and LN $\in \{1,2,3,4\}$) and then for the two marginal LN classes LN $\geq 3$ and LN = 1, merging all RB classes (poor, fair, good). As the decrease of <$P_{unstable}$> and the proportion of profiles classified as unstable was more pronounced from the LN $\geq 3$ to the LN=1 subset than from the "poor" to the "good" RB class, we concluded that the simulated stability correlated more strongly with the local danger level estimate (LN) than with the observed stability at a point as assessed with a RB test. We rewrote the paragraph in the revised manuscript to improve clarity.

L389-390: could you plot on Fig. 13 the avalanche and non-avalanche days as defined in this paper.

We regret but do not understand what you ask us to do here.

L402: « Figure c » => « Figure 15 c » ?

Changed as suggested.

L425: « they were mostly developed to align complete profiles ». In practice, this is not true as a parameter of the model can be used to align only a sub part of the profile. In particular it is used to relax the assumption that the snow-ground interface must be matched. Besides, it is not a limit of the method since for the manually matching you also look below the weak layer for stratigraphy markers (eg. MF-crust). « these additional parameters are not included in the current available automated methods » It is implemented and shown in Viallon-Galinier et al. (2020). Actually your manual method seems to works fine enough and you do not necessarily need an automatic method. You might see the automated matching method as a further development to reduce the time spent to prepare the data but you do not need to say something wrong about the automated method limits.

We rewrote the paragraph and included the reference you suggested in the revised manuscript.

Section 5.3 : all your analysis is based on the feature importance as computed by the scipy package. First, here, you do not give any information on the « sign » (> or <) of the important feature. For instance, it is not clear (and there is no info about that) whether it is high or low values of « mean density divide by mean grain size » that promote instability. To be added. Besides, the feature importance are somehow « shared » between correlated variables. For instance, viscous deformation might be correlated to the initiation criteria such as SK38 (stress over strength) which is itself correlated to strength, stress (and so importance shared ...). Your comment about the absence of initiation criterion must therefore be qualified. Moreover, your comparison of your model score (6 parameters, training) to the « physical » model with only two parameters and no training is unfair (L. 478).

Thank you for your recommendations on how to improve Section 5.3. To include information on the "sign" of the relationship between the features and the target response, we included partial dependence plots in the appendix of the revised manuscript. A partial dependence plot shows the effect of a given feature on the output prediction, marginalizing over the values of all other features (Friedmann, 2001).

While training the RF model we aimed at avoiding "shared" feature importance between correlated features by excluding pairs of features that were highly correlated (Pearson r > 0.8). The correlation between viscous deformation rate and the skier stability index SK38 in our training data set was rather low (Pearson r =- 0.19). Even when removing all features with correlation coefficients with SK38 exceeding 0.5, SK38 still appears at the lower end of the feature importance ranking.

We agree that the comparison of our trained model with the untrained threshold-based model using only the critical crack length and the initiation criterion as input features is somewhat unfair. In the revised

manuscript, we made the limitations of this comparison more explicit. We also noted that when training a decision tree of depth two on the DAV data set, the five-fold cross-validated accuracy was lower when using the critical crack length and the failure initiation criterion as compared to using only the critical crack length. This clearly indicates that for our data set the strength-over-stress initiation criteria have very limited added-value.

Fig. 13 and 14 and Section 4.2.5: the results at the regional scale are very interesting but never discussed in the paper. In particular, the model apparently failed (?) to detect clearly the big avalanche events (high AAI) at the regional scale. Add a discussion on the inherent difficulty to predict high AAI from only slab stability indices (size, spatial distribution, natural release).

We agree that there are some discrepancies between the predictions of our RF classifier and the observed regional avalanche activity and that we did not mention these results in the discussion. Our main goal with these two figures was to show the potential applicability of our RF classifier for avalanche forecasting and the overall promising results. As we only used simulations from one field site for this comparison, there can be a number of reasons why these discrepancies occur, including a lack of information on spatial snow distribution and on potential avalanche size as well as incomplete or biased avalanche data. As these are well-known problems when using avalanche observations for validation, we briefly discussed the results shown in these figures in section 5.5, but did not discuss these potential error sources in great length.

**Reply to Reviewer #2: Edward Bair**

At 27 pages with 15 figures and 2 tables, excluding the 2 appendices, the article is too long. The Cryosphere is unusually vague in article size limits, but it is expected to fit with 12 journal pages. In any case, the article's length dilutes its important findings, which show that random forests can be used to classify profiles based on stability with high accuracy. Perhaps some of the details regarding hyperparameters and explanation of the widely-used random forest model could be omitted or moved to an appendix.

While revising the manuscript, we shortened the text wherever we deemed it feasible.
In general, we prefer a comprehensive rather than a brief presentation for reasons of transparency and reproducibility.

The finding that viscous deformation is the most important predictor is only briefly discussed. This finding deserves further discussion as it highlights how profiles alone are inadequate to classify instability. Loading rate is one of the most important avalanche predictors, stated in Atwater and Koziol (1953) and before. The viscous deformation parameter appears to be an indirect measure of this.

We looked in more detail at the equations which define the viscous deformation rate. To model the settling of the snowpack, SNOWPACK treats snow as a viscoelastic material which accounts for elastic as well as nonelastic irreversible deformations following the constitutive equations of a Maxwell model (Bartelt and Moos, 2000; Bartelt and Lehning, 2002). The total strain $\varepsilon$ is thus composed of an instantaneous elastic part $\varepsilon_e$ and a time dependent viscous part $\varepsilon_v$, i.e. $\varepsilon = \varepsilon_e + \varepsilon_v$. While the elastic strain rate is directly related to the loading rate $\dot{\sigma}_n$ via $\dot{\varepsilon}_e = \frac{\dot{\sigma}_n}{E}$, where E is the elastic modulus, the viscous deformation rate is related to the normal stress via $\dot{\varepsilon}_v := \frac{\sigma_n}{\nu}$ where $\nu$ is the viscosity. There is thus no direct link between the

viscous deformation rate and the loading rate. Nevertheless, we would expect the loading rate, if included into the input features, to appear at the top of the feature importance ranking, as it is directly related to the amount of new snow, the strongest forecasting parameter for large avalanches (Schweizer et al., 2003). Indeed, the amount of new snow was the most important parameter for other instability or avalanche danger models (Schirmer et al., 2010, Perez et al., 2022). For our RF model, we however intentionally excluded meteorological parameters, such as the amount of new snow, to avoid blurring the importance of parameters describing snow stratigraphy.

L62 citation?

We moved the citation for the RF classification (Breiman, 2001a).

L220 put this citation at the first mention of RF on l 62. Since RF is already defined there, "Random Forest" need not be spelled out here.

Changed as suggested.

L222 delete accounting

We replaced the wording by: ..., this model can account for complex mutual dependencies ...

L412 linked

Changed as suggested.

L526 detection of

We replaced "detecting" with "the detection of" in the revised manuscript.

**References:**

Bartelt, P. and Lehning, M.: A physical SNOWPACK model for the Swiss avalanche warning Part I: Numerical model, Cold Reg. Sci. Technol., 35, 123–145, https://doi.org/10.1016/S0165-232X(02)00074-5, 2002.

Bartelt, P., and Moos, M.V.: Triaxial tests to determine a microstructure-based snow viscosity law, Annals of Glaciology, v. 31, 457–462, doi: 10.3189/172756400781819761, 2000.

Brenner, H. and Gefeller, O.: Variations of sensitivity, specificity, likelihood ratios and predictive values with disease prevalence, Stat. Med., 16, 981–991, https://doi.org/10.1002/(SICI)1097-0258(19970515)16:9<981::AID-SIM510>3.0.CO;2-N, 1997

Friedman, J.H.: Greedy function approximation: A gradient boosting machine, The Annals of Statistics, 29(5), 1189-1232, https://doi.org/10.1214/aos/1013203451, 2001.

Giraud, G.: MEPRA: an expert system for avalanche risk forecasting, Proceedings ISSW 1992. International Snow Science Workshop, Breckenridge, Colorado, U.S.A., 4-8 October 1992, pp. 97-106, 1993.

Pérez-Guillén, C., Techel, F., Hendrick, M., Volpi, M., van Herwijnen, A., Olevski, T., Obozinski, G., Pérez-Cruz, F., and Schweizer, J.: Data-driven automated predictions of the avalanche danger level for dry-snow conditions in Switzerland, Nat. Hazards Earth Syst. Sci., 22, 2031–2056, https://doi.org/10.5194/nhess-22-2031-2022, 2022.

Schweizer, J., Jamieson, J. B., and Schneebeli, M.: Snow avalanche formation, Rev. Geophys., 41, 1016, https://doi.org/10.1029/2002RG000123, 2003.

Schweizer, J. and Jamieson, B.: Snowpack tests for assessing snow-slope stability, Ann. Glaciol., 51, 187–194, https://doi.org/10.3189/172756410791386652, 2010.

Schirmer, M., Schweizer, J., and Lehning, M.: Statistical evaluation of local to regional snowpack stability using simulated snow-cover data, Cold Reg. Sci. Technol., 64, 110–118, https://doi.org/10.1016/j.coldregions.2010.04.012, 2010.

Techel, F., Winkler, K., Walcher, M., van Herwijnen, A., and Schweizer, J.: On snow stability interpretation of extended column test results, Nat. Hazards Earth Syst. Sci., 20, 1941–1953, https://doi.org/10.5194/nhess-20-1941-2020, 2020a.

Techel, F., Müller, K., and Schweizer, J.: On the importance of snowpack stability, the frequency distribution of snowpack stability, and avalanche size in assessing the avalanche danger level, The Cryosphere, 14, 3503–3521, https://doi.org/10.5194/tc-14-3503-2020, 2020b.

Viallon-Galinier, L., Hagenmuller, P., Lafaysse, M.: Forcing and evaluating detailed snow cover models with stratigraphy observations. Cold Regions Science and Technology 180, 103163, https://doi.org/10.1016/j.coldregions.2020.103163, 2020.